# A single-cell atlas to map sex-specific gene-expression changes in blood upon neurodegeneration

Friederike Grandke [1], Tobias Fehlmann[1], Fabian Kern [1,2], David M. Gate [3,4,5,6], Tobias William Wolff [1], Olivia Leventhal[3], Divya Channappa[3], Pascal Hirsch [1], Edward N. Wilson [3], Eckart Meese[7], Chuanyu Liu[8], Quan Shi[8], Matthias Flotho [1], Yongping Li[1,8], Cynthia Chen[8], Yeya Yu[8], Jiangshan Xu[8], Michael Junkin[8], Zhifeng Wang[8], Tao Wu[8], Longqi Liu[8], Yong Hou[8], Katrin I. Andreasson [3,5,9], Jenny S. Gansen[1], Elvira Mass [10], Kathleen Poston [3], Tony Wyss-Coray [3,4,5,6,11,13] ✉ & Andreas Keller [1,2,3,12,13] ✉

The clinical course and treatment of neurodegenerative disease are complicated by immune-system interference and chronic inflammatory processes, which remain incompletely understood. Mapping immune signatures in larger human cohorts through single-cell gene expression profiling supports our understanding of observed peripheral changes in neurodegeneration. Here, we employ single-cell gene expression profiling of over 909k peripheral blood mononuclear cells (PBMCs) from 121 healthy individuals, 48 patients with mild cognitive impairment (MCI), 46 with Parkinson's disease (PD), 27 with Alzheimer's disease (AD), and 15 with both PD and MCI. The dataset is interactively accessible through a freely available website (https://www.ccb.uni-saarland.de/adrcsc). In this work, we identify disease-associated changes in blood cell type composition and the gene expression in a sex-specific manner, offering insights into peripheral and solid tissue signatures in AD and PD.

Neurodegenerative disorders such as Alzheimer's disease (AD) and Parkinson's disease (PD) show increasing rates of prevalence in the global population and severely affect the quality of life of the elderly[1,2]. Common risk factors include the genetic background, personal lifestyle, environmental exposure and age[3]. Although the clinical manifestation, diagnosis and course of progression vary substantially between AD and PD, underlying common and converging cellular

pathways have been proposed[4]. Thereby, both diseases can arise from distinct precursor stages of varying duration and co-occur with certain shared symptoms, including (mild) cognitive decline[5,6]. Notably, the interplay of the adaptive immune system and brain inflammation in neurodegenerative diseases is increasingly appreciated[7,8]. In this context, substantial efforts have carved out the primary affected central nervous system (CNS) cell types and key risk genes driving the

[1]Clinical Bioinformatics, Saarland University, 66123 Saarbrücken, Germany. [2]Helmholtz Institute for Pharmaceutical Research Saarland (HIPS)–Helmholtz Centre for Infection Re- search (HZI), Saarland University Campus, Saarbrücken, Germany. [3]Department of Neurology and Neurological Sciences, Stanford University, Stanford, CA 94305, USA. [4]Veterans Administration Palo Alto Healthcare System, Palo Alto, CA, USA. [5]Wu Tsai Neurosciences Institute, Stanford University, Stanford, CA, USA. [6]Chemistry, Engineering, and Medicine for Human Health, Stanford University, Stanford, CA, USA. [7]Department of Human Genetics, Saarland University, 66421 Homburg/Saar, Germany. [8]MGI Group, San Jose, CA, USA. [9]Program in Immunology, Stanford University, Stanford, CA, USA. [10]Life and Medical Sciences Institute, Developmental Biology of the Immune System, University of Bonn, Bonn, Germany. [11]The Phil and Penny Knight Initiative for Brain Resilience, Stanford University, Stanford, CA, USA. [12]PharmaScienceHub, Saarland University Campus, Saarbrücken, Germany. [13]These authors jointly supervised this work: Tony Wyss-Coray, Andreas Keller. ✉e-mail: twc@stanford.edu; andreas.keller@ccb.uni-saarland.de

underlying pathology, e.g., *APOE* and *TREM2* in microglia for AD, or alpha-synuclein (*SNCA*) contained in Lewy-bodies in dopaminergic neurons for PD[9–13]. In a similar direction, single cell technologies facilitate to uncover the role of the human brain vasculature for the development of neurodegenerative disorders[14,15].

While the brain regions selectively affected in these diseases are subject to intense but predominantly small-scale research due to the apparent difficulties to obtain fresh-frozen human tissue samples and associated costs, research has focused on model organisms e.g., genetically modified mice, in vitro cell systems, or on the in human better accessible peripheral organic system, for example blood or cerebrospinal fluid (CSF) and their connecting interfaces such as the blood-brain barrier (BBB)[16–18]. Indeed, accumulating evidence points to a partial breakdown of the BBB with age and during age-related diseases, suggesting a still underestimated role of the peripheral immune system in neurodegenerative conditions, potentially through information exchange between these otherwise restrictively isolated physiological environments[19–22]. These findings raise questions about how and at which point during the disease progression the peripheral immune system is implicated, especially in the context of driving systemic inflammation[23–26]. One key factor which is increasingly considered in the pathology of AD and PD is the patients' biological sex. Single-cell studies for AD for instance suggest that female cells are overrepresented in disease-associated subpopulations and that transcriptional responses are different between sexes in oligodendrocytes and other cell types[11]. Sex-dependent changes in brain and the blood of AD patients have been repeatedly reported (Supplementary Table 1). In contrast, sex-dependent changes in PD patients in the blood have so far only been reported in monocytes[27].

Broadly accessible and low-invasively collectable plasma samples offer great promise for the detection of circulating biomarkers in neurodegeneration through a screening of peripheral blood mononuclear cells (PBMC), especially in large-scale settings[28–32]. Substantial efforts on clinical cohorts showed that plasma markers measured with analytical methods can be used to partially predict or track neurodegenerative disease development while bearing also its own challenges[33–37]. Thus, a next logical step is to comprehensively map molecular pathways correlating with, or even causative for these disease biomarker signatures using high-throughput techniques at high resolution. Single-cell RNA sequencing (scRNA-seq) holds the promise of unraveling cell type-specific homeostatic conditions and their alterations in human disease, as recently demonstrated for parenchymal brain tissue, brain vasculature, or PBMCs in neurodegenerative diseases[11,12,14,15,38–44]. Reaching numbers of up to one million cellular gene expression profiles per publication, the amount of transcriptomic data generated ranges on exponential scales and is expected to further increase[45–48].

In this work, we sought to characterize the peripheral response through the transcriptional landscape[49] in neurodegeneration by profiling over 909k PBMCs of 290 blood samples including 155 samples of individuals with clinical neurodegeneration compared to 135 samples of healthy controls (Fig. 1a, Supplementary Data 1). We thus surpass all other currently published datasets of RNA-sequencing data of PBMCs in Alzheimer's and Parkinson's disease in both the number of patients and the number of cells (Supplementary Fig. 1, Supplementary Table 2). Our dataset offers insights into the general sensing of strongly CNS-associated diseases in the periphery, for the first time evaluating the potential of low-invasive cellular biomarkers for these, both at single-cell resolution and at scale. By including time-series follow-ups for a subset of the patients, as well as additional gray and white matter volume measurements and protein-levels of known AD-disease markers in the CSF, we combine a variety of patient-centric data. A companion web browser is available at https://www.ccb.uni-saarland.de/adrcsc and enables researchers to easily access the pre-processed dataset without requiring technical expertise and simplifies the testing of data-driven hypotheses.

## Results

### A scRNA-seq dataset of PBMCs in neurodegenerative diseases
From the Stanford Alzheimer's Disease Research Center (ADRC) we selected patients diagnosed with mild cognitive impairment (individual patients n = 48, total samples from the patients $n_t$ = 55; MCI), Alzheimer's disease (n = 27, $n_t$ = 34; AD), Parkinson's disease (n = 46, $n_t$ = 48; PD), PD with MCI ($n$ = 15, $n_t$ = 18; PD-MCI), and healthy individuals ($n$ = 121, $n_t$ = 135; HC) (Fig. 1b, Table 1). We ensured a well-balanced ratio of both biological sex and age between the subgroups (Supplementary Fig. 2a). For each sample, we performed droplet-based single-cell RNA-sequencing[49] on collected PBMCs, totaling 1,374,714 cellular expression profiles. We then kept the top 909,322 single cells of highest quality (66%) that passed stringent quality control filtering (Supplementary Fig. 2b). Following a cluster-based cell type annotation and manual marker curation, we found 13 major PBMC types that were further divided into 33 fine-granular sub-cell types (Fig. 1c, d, Supplementary Fig. 3). The downstream analysis described in this paper was performed using only the baseline sample of each patient (unless stated otherwise).

### Sex-specific changes in cell type proportions
As visualizing those large cell numbers in a 2-dimensional embedding becomes unclear - a phenomenon commonly denoted as overplotting - we estimated the density of cells to highlight the distribution of cells for each diagnosis group (Supplementary Fig. 4a). Interpreting the projected densities, we found slight differences in the distribution of cells, suggesting a disease-specific transcriptomics shift in the overall abundance of different cell type populations. In light of previous reports indicating that changes in cell type proportions in PBMC samples may differ between males and females[50], we separated the data by sex to independently examine disease-related changes. In general, we observed distinct changes in the cell type proportions in men and women (Supplementary Fig. 4b). A density analysis depicts these effects in the integrated data space (Fig. 2a).

These sex-dependent changes were confirmed using scCODA, which did not show significant changes in the cell-type distributions (abs log2FC > 0.35, significant according to scCODA) when leaving out sex as a covariate (Supplementary Fig. 4c). Split by male and female patients, we were able to observe significant changes (Fig. 2b), some of which were different dependent on the sex. In Parkinson's disease, CD8 + T cells and Plasma cells show a positive fold-change in males but a negative fold-change in females. Similarly, B cells are more abundant in females with PD-MCI but less abundant in males with PD-MCI. Although a sex-dependent change in B and NK cells of AD patients was indicated in the density embedding, scCODA did not show a significant difference. In the finer cell-annotation, we observed significant changes in only 3 cases in males and 9 cases in females using the raw cell type proportions (Fig. 2c). The significant changes in AD, MCI and PD match the ones found in the female patients (Supplementary Fig. 4d). Using scCODA on the finer Annotation, we found a lower proportion of $T_{fh}$ cells in AD, as previously reported (Fig. 2d). When comparing these findings with previously reported changes, some were previously described by others, such as the increased proportion of Monocytes in AD, and changes in B cells in PD (Fig. 2e).

Several but not all of the changes we report in this study have been previously described by others, such as the increased proportion of B cells in AD, and changes in CD8 + T cells in PD. Overall, our results suggest a significant shift in the cellular composition of male and female patients, with disease-specific differences. An important question to consider is whether these changes are driven by small, distinct sets of genes, or broader transcriptomic perturbations.

### Differences in the sex-specific transcriptome
To gain further insight into the respective molecular changes among male and female patients with MCI, AD, or PD, we conducted a DEG analysis (Supplementary Data 2). As the sample sizes of PD-MCI is too

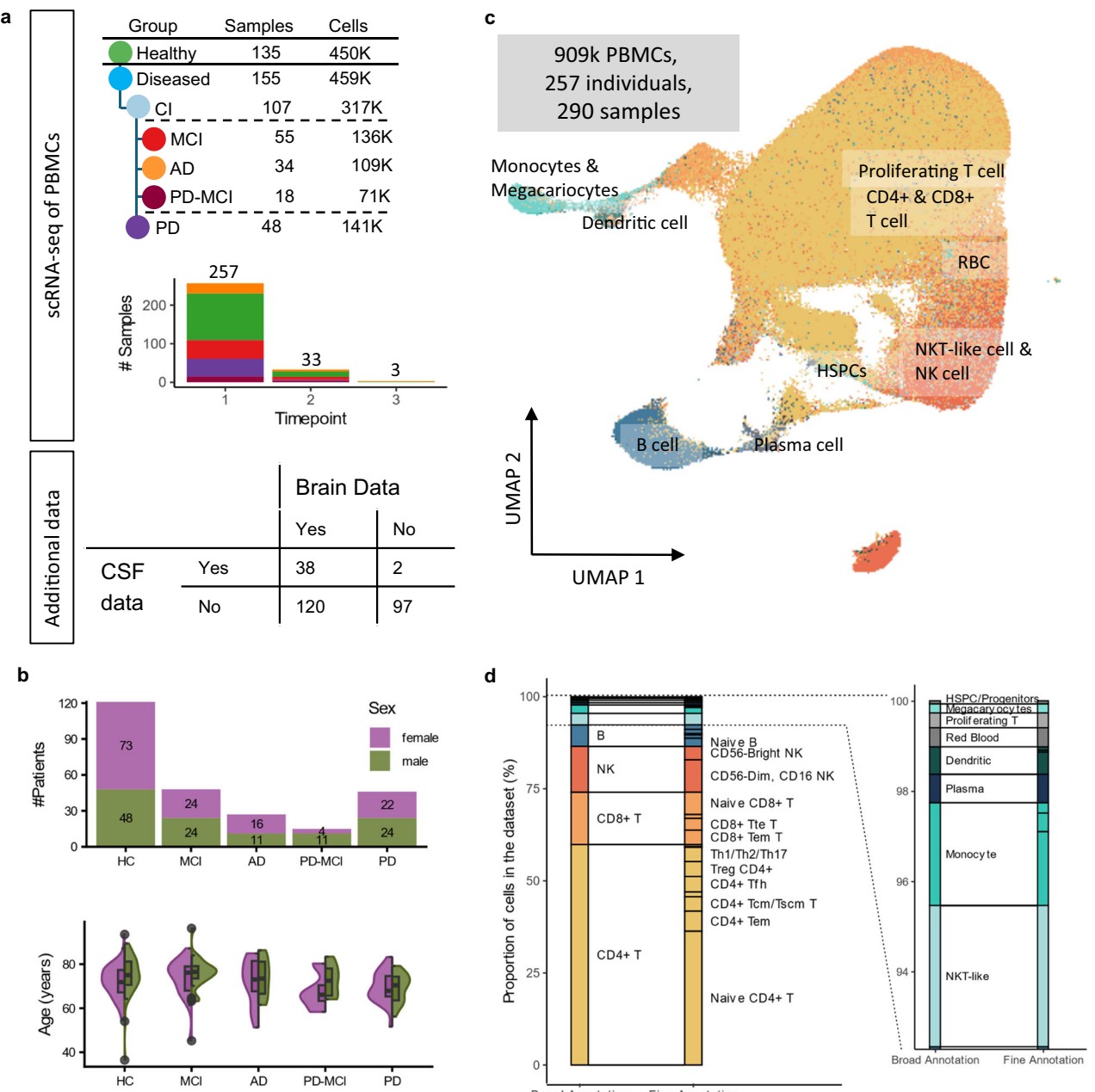

**Fig. 1 | Cohort and single-cell data characterization. a** Overview on the dataset. PBMC samples were collected from healthy patients (HC), patients with Parkinson's Disease (PD), and patients with Cognitive Impairment (CI), more detailed with Mild Cognitive Impairment (MCI), Alzheimer's Disease (AD), and Parkinson's Disease with Mild Cognitive Impairment (PD-MCI). For a subset of patients, blood samples were drawn at multiple time-points. The samples were then profiled using single-cell RNA sequencing. For a subset of patients, additional data was collected on brain volumes (158 patients) and known Alzheimer biomarkers were measured in the CSF (40 patients). **b** Distribution of biological sex and age per patient group (divided by sex). The violin plots show the shape of the distribution, while the boxes encompass the first through third quartiles, with the central line marking the median. The whiskers mark the minimum or maximum values or, if outliers are present, the values within 1.5 times the interquartile range of the first or third quartile. Outliers are shown as dots. **c** Two-dimensional representation of the $n = 909$k cells using Uniform Manifold Approximation and Projection (UMAP). Cells are colored by assigned cell type. **d** Cells are annotated on different levels. This allows the analysis on a broad level with 13 bigger clusters or on a finer level with 33 clusters.

small for most analysis we excluded it from further in-depth consideration. Our analysis aims to identify both, sex- and disease-related differences in gene expression. We revealed a stark contrast between sexes in the number of DEGs, particularly in Alzheimer's patients, which has been previously noted by Mathys et al. [11] (Fig. 3a). Specifically, we detected de-regulated genes in only 12 cell types in male patients, whereas in female patients 20 cell types exhibited de-regulated genes. A similar, albeit less sex-pronounced trend was observed in the comparison between PD and HC, as well as MCI vs. HC patients. We then compared the fold-changes in male and female subgroups and found a positive correlation for MCI vs.

HC (overall correlation of 0.35, p-value < $2.2 * 10^{-16}$) and AD vs. HC (overall correlation of 0.32, p-value < $2.2 * 10^{-16}$), particularly in cell types with a high number of significantly de-regulated genes (Fig. 3b). Conversely, cell types in PD vs. HC exhibited lower or no correlation at all that are often negative (overall correlation = −0.0064, p-value = 0.098). Overall, our analysis revealed that most of the cell types in the PD vs. HC comparison were either negatively (9 cell types) correlated or not significantly correlated while most cell types exhibited a positive correlation in both AD (22 cell types) and MCI (22 cell types) (Fig. 3c). In total, we identified only 84 genes that were de-regulated independent of the sex.

**Table 1 | Demographic breakdown of the patients included in this study**

| | | Age in females (mean, sd and number of patients) | | | Age in males (mean, sd and number of patients) | | | Adj. P-value (m vs. f) | | |
|---|---|---|---|---|---|---|---|---|---|---|
| Diagnosis | HC | 72 ± 7 (n = 82) | | | 75 ± 9 (n = 53) | | | 0.318 | | |
| | AD | 71 ± 10 (n = 21) | | | 73 ± 8 (n = 13) | | | 1 | | |
| | PD | 69 ± 7 (n = 23) | | | 69 ± 6 (n = 25) | | | 1 | | |
| | MCI | 73 ± 8 (n = 29) | | | 76 ± 6 (n = 26) | | | 1 | | |
| | PD-MCI | 68 ± 7 (n = 6) | | | 73 ± 6 (n = 12) | | | 1 | | |
| Race | White (w) | 71 ± 8 (n = 144) | | | 74 ± 7 (n = 116) | | | 0.101 | | |
| | Asian (a) | 72 ± 7 (n = 12) | | | 72 ± 14 (n = 9) | | | 1 | | |
| | Other (o) | 73 ± 7 (n = 5) | | | 74 ± 7 (n = 4) | | | 1 | | |
| | Race | w | a | o | w | a | o | w | a | o |
| Diagnosis | HC | 72 ± 7 (n = 72) | 71 ± 6 (n = 6) | 71 ± 6 (n = 4) | 76 ± 7 (n = 49) | 64 ± 20 (n = 4) | - (n = 0) | 0.148 | 1 | - |
| | AD | 70 ± 10 (n = 18) | 75 ± 8 (n = 3) | - (n = 0) | 73 ± 8 (n = 13) | - (n = 0) | - (n = 0) | 1 | - | - |
| | PD | 69 ± 7 (n = 22) | 67 - (n = 1) | - (n = 0) | 68 ± 6 (n = 23) | 73 ± 4 (n = 2) | - (n = 0) | 1 | - | - |
| | MCI | 73 ± 8 (n = 26) | 69 ± 11 (n = 2) | 83 - (n = 1) | 75 ± 6 (n = 20) | 83 ± 1 (n = 2) | 74 ± 7 (n = 4) | 1 | - | - |
| | PD-MCI | 68 ± 7 (n = 6) | - (n = 0) | - (n = 0) | 72 ± 7 (n = 11) | 78 - (n = 1) | - (n = 0) | 1 | - | - |

More detailed information can be found in Supplementary Data 1.

Most of these genes were found in CD4+ $T_{cm}/T_{scm}$ T cells, NKT-like cells (Fig. 3d) or in CD56-Dim, CD16 NK cell. These findings lead us to evaluate if the similarities between the different diseases varies by sex. We thus determined the overlap between the significantly de-regulated genes in the three diseases to control comparisons. The distribution of shared genes between diseases in males (Fig. 3e) compared to those in females (Fig. 3f) is similar and significantly de-regulated genes are mainly shared between MCI vs. HC and PD vs. HC. In particular, 35 genes are differentially expressed in all three comparisons for male and 27 for female patients. Overall, our results thus suggest rather sex-independent expression patterns in the different diseases, while in certain cases - depending on the disease and cell type - the signatures for the sexes differ substantially. When comparing the gene-expression changes in the different cell-types, we observed that the changes in sub-cell types from the same cell type show a high similarity (Supplementary Fig. 5).

## Peripheral pathways affected in neurodegeneration

In the light of the significant variability in gene expression patterns among diseases, cell types, and sexes, we sought to verify that signatures in our dataset reflect the dysregulation of known disease gene markers. To accomplish this, we specifically analyzed genes from the KEGG Alzheimer's disease and Parkinson's disease pathways that were de-regulated in our PBMCs. Astonishingly, we found that gene expression in the KEGG Alzheimer's disease pathway is largely independent of both sex and cell type. This might be due to a bias in how the genes on the pathways are identified in previous studies, largely excluding sex specific effects. While the fold-changes in some cell types differ between sexes, the direction of the de-regulation is consistent (Supplementary Fig. 6a). Conversely, we observed differences in gene expression between males and females and between different cell types for some genes in the KEGG Parkinson's disease pathway (Supplementary Fig. 6b).

This poses the question, which pathways are affected specifically in males and females. The results of a pathway analysis of the change in all cell-types and sexes (Supplementary Data 3) shows that the affected pathways in AD and in MCI are similar in both sexes (Fig. 4a). In PD the pathways are more sex specific. The overall most frequently enriched pathway in AD is "SRP-dependent cotranslational protein targeting to membrane". This is also the most frequently enriched pathway in females MCI and males with PD (Fig. 4b). In PD, the "Asthma" pathway is most affected in females and in MCI "Oxidative phosphorylation" in males.

In PD, MCI and in males in AD, the pathways are similar across all cell-types (Supplementary Fig. 7). In females with AD, B cells show a similar cluster of pathways consisting of pathways related to junctions (adherens junction, cell-substrate junction, cell-substrate adherens junction) and pathways related to catabolic and biosynthetic processes ("amide biosynthetic process", "aromatic compound catabolic process",...). In males, these pathways appear across all groups of cell-types.

Overall, there are 15 pathways shared between PD, MCI and AD in females, but 53 in males (Fig. 4c, d). 114 pathways are shared between AD and PD in males and 24 in females.

## Similar disease signatures in PBMCs and neural cells

Having determined changes that can be observed between Alzheimer's disease and healthy controls in PBMC samples, we sought to contrast peripheral cell transcriptomes to those from the brain. A large number of studies describe single cell gene expression across multiple regions in the brain with respect to different disease phenotypes and physiological conditions, complicating a joint analysis.

We thus used a database that integrates single cell gene expression data from 21 studies, allowing for precise insights into the regulation of genes across cell types in relation to age, sex, and disease (Fig. 5a)[51]. As there are not enough cells from the same cell-type in both brain and blood, we rely on overlaps of gene-expression patterns and pathways when looking for similar patterns in neurodegenerative diseases.

We identified genes that exhibit differential expression patterns between Alzheimer's disease and healthy individuals in male and female patients for the PBMCs and investigated whether these exhibit similar expression patterns in brain cells of Alzheimer's patients (Supplementary Data 4, Supplementary Fig. 8a). Our findings suggest that gene expression changes in peripheral cells may not necessarily reflect those in the brain, given the natural disparity of cell types between blood and brain. However, emphasizing the need for further in-depth investigation knowing the role of the immune system in Alzheimer's, the associated brain inflammation and infiltration by specifically activated immune cells.

Overall, there were 36 genes in males and 7 genes in females that were significantly de-regulated in both PBMCs and the brain in the main cell-types (Fig. 5b, c). The genes in males (adj. P-value < 0.05 and abs. log2 Fold-change > 0.5 in both datasets) were enriched for pathways related to the regulation of the immune system and the membrane (Supplementary Fig. 8b, c). The female genes (adj. P-value < 0.05 in both datasets) were enriched for the Herpes Simplex Virus 1 pathway. Herpes simplex has previously been linked to an increased risk for the development of Alzheimer's and is discussed as a key factor in disease development[52,53].

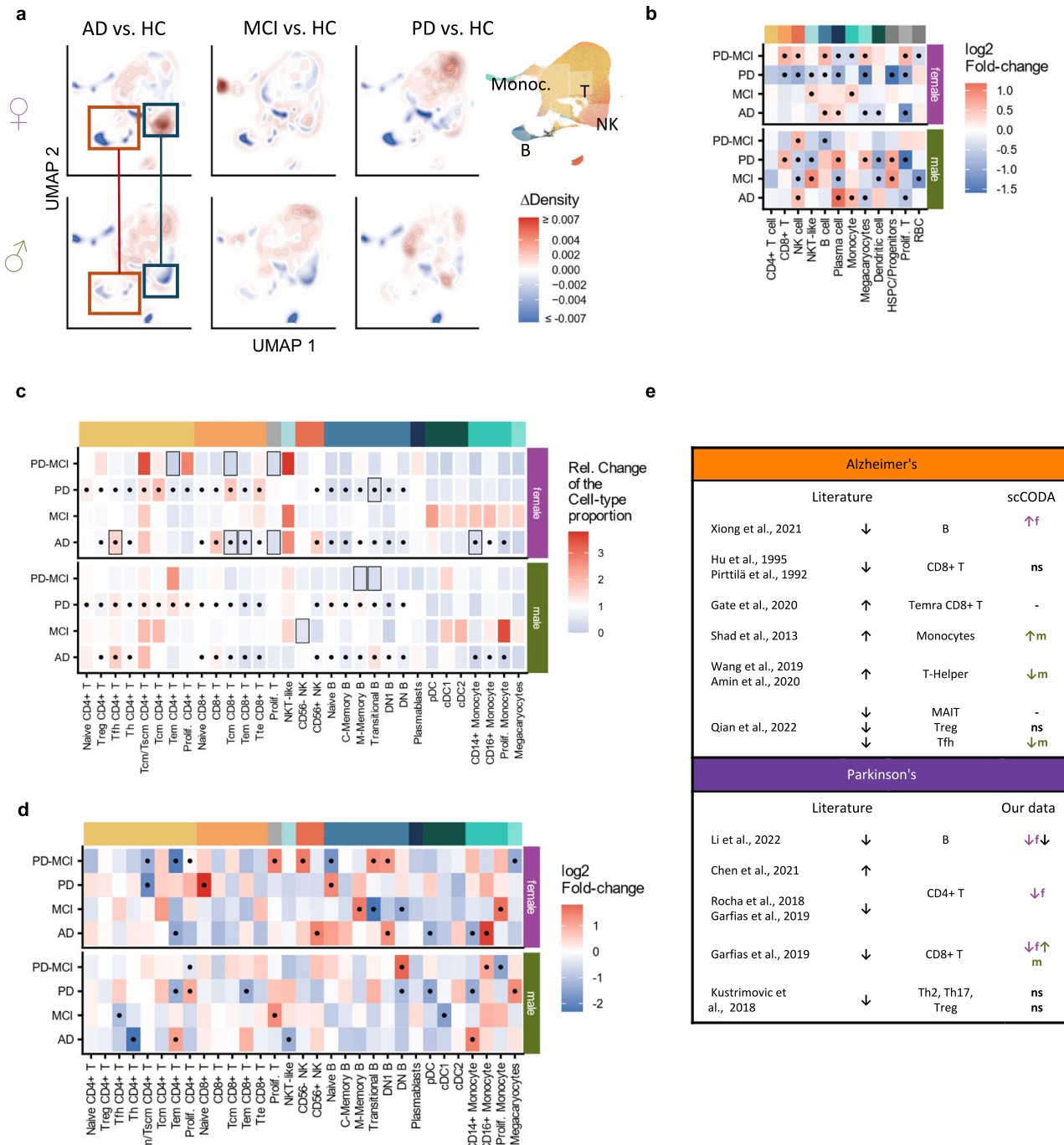

**Fig. 2 | Sex-dependent changes of cell type proportions. a** Differential density UMAP embedding for male and female patients shows differences in the cell type distribution for AD, PD and MCI. Each embedding shows the spots with the highest density of the diseased patient group (red) or the healthy control group (blue). **b** Comparison of the cell type proportions (on the broad annotation) using scCODA with Age and ApoE as covariates. Celltypes with an absolute Fold-change of at least 0.5 that were reported as significant by scCODA are marked with a dot. **c** Relative change in the cell type-proportion in % between Healthy and the diagnosis groups for the different cell types reveals sex-specific changes. Significant values (adj. p-value < 0.05, unpaired Student's t-test) are marked by a frame, previously described changes from the literature are marked with a dot. **d** Comparison of the changes of cell type proportions found in the literature and the cell type proportion changes found in our PBMC data using scCODA. Changes are only considered if they were reported as significant by scCODA between healthy and disease. If no sex is given, the test was performed without considering the sex. **e** Comparison of the cell type proportions on the fine Annotation using scCODA as described in (**b**). Celltypes with an absolute Fold-change of at least 1 that were reported as significant by scCODA are marked with a dot.

To further validate these findings and to see if the expression changes with different brain-regions, we tested for the general overlap of differentially expressed genes (abs. log2 Fold-change > 0.5 and adj. P-value < 0.05) in this PBMC dataset, the ZEBRA dataset (21 studies) and an additional ROSMAP dataset (4 studies; 6 brain regions) (Supplementary Data 5). Of note, two studies from the prefrontal cortex samples of the ROSMAP dataset are a subset of the ZEBRA dataset and thus not independent. The other brain-regions are independent between the datasets. We found an overlap of 32 genes in the male patients and 8 genes in the female patients. 24 out of the 32 male genes and 5 out of the 8 female genes have been previously reported in the context of Alzheimer's disease, most in the context of the immune

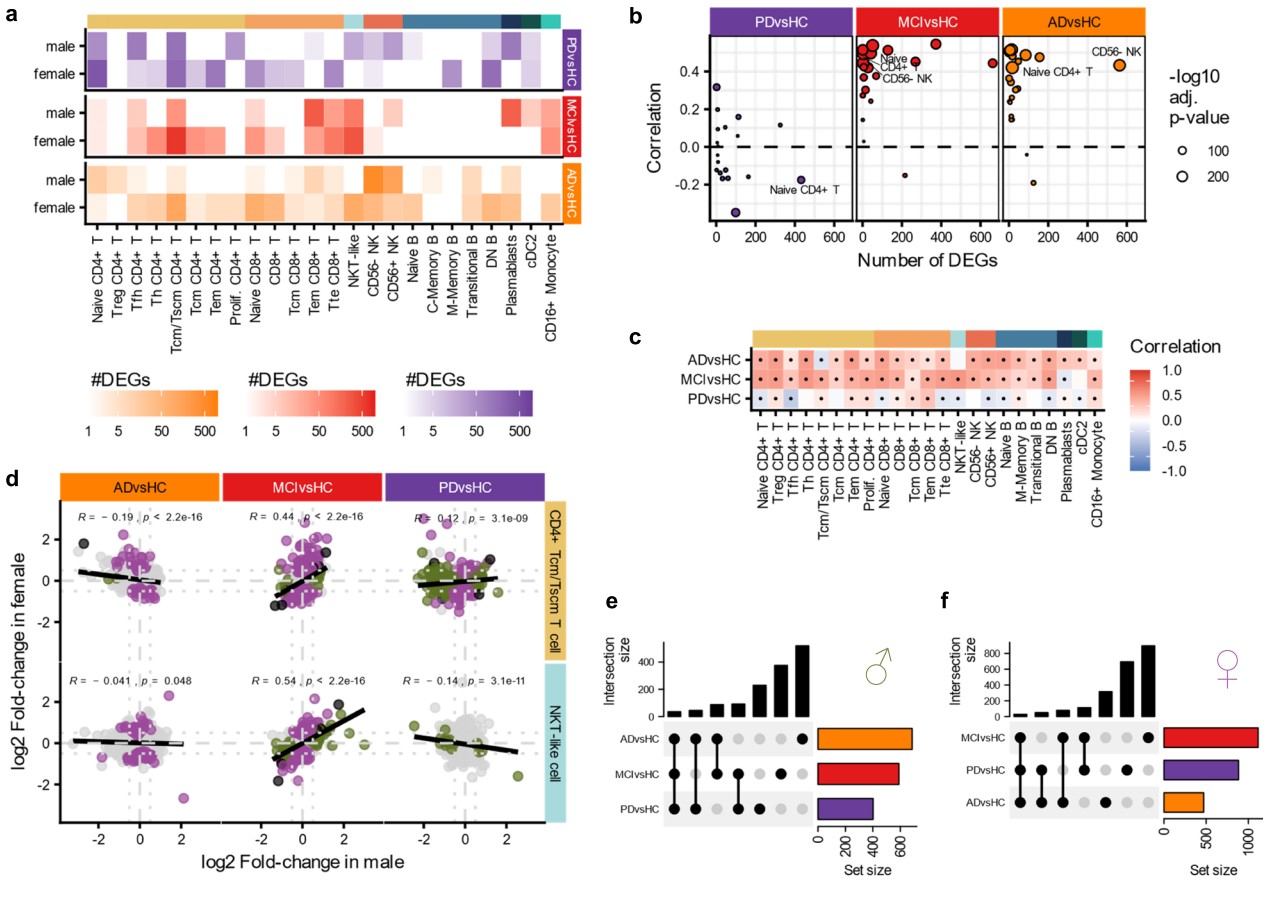

**Fig. 3 | Changes in gene-expression patterns in male and female patients.**
**a** Comparison of the number of genes that are significantly de-regulated (adj. p-value < 0.05 and absolute log2 fold-change > 0.5, see Methods) between the healthy and diseased patients in females and in males. **b** Cell types with more significantly de-regulated genes (as in a) show higher correlation values in AD and MCI. The Pearson correlation coefficient was calculated between the fold-changes in male and female of all genes in the dataset. **c** Correlations of the fold-changes in male and female patients in PD is often lower and in 9 cell types negative, but positive in most cell types in MCI and AD. Significant values

(Pearson's correlation, adj. p-value < 0.05) are marked with a dot. **d** Comparison of the log2 Fold-changes in male and female patients in CD4+ $T_{cm}$/$T_{scm}$ cells and NKT-like cells shows the number of genes that are significantly de-regulated in both or only one sex. The genes are colored by the sex in which they are significant in (adj. p-value < 0.05, see Methods). **e, f** Upset plot showing the number of significantly de-regulated genes that are shared between the different comparisons of healthy with the diagnosis groups for the male sub-group (e) and for the female sub-group (f) show similar patterns in the overlap between diseases in both sexes.

system, blood-brain barrier, Astrocytes and Microglia (Supplementary Table 3, Fig. 5d, e).

To gain a deeper understanding for the changes in PBMCs and brain cells and how they might interact, we compared the de-regulated genes with the CellChat signaling database[54]. We found genes from the *CCL* (Supplementary Fig. 8d) and the *MHC-I* signaling pathway in the de-regulated genes in females and the *CD45* and the *CCL* signaling pathway in males. All of these genes have already been reported in the context of Alzheimer's in the literature[55–57]. This indicates that changes in the blood might influence the brain or vice versa, but as there are only few genes of these pathways de-regulated, further experiments would be necessary to confirm this.

Additionally, we performed a pathway analysis using both the de-regulated genes in male and female samples (Supplementary Data 6 and 7). We observed a general enrichment of membrane, ribosome, and adherens junction-related pathways in both blood and brain (Fig. 5f). Especially the "SRP-dependent cotranslational protein targeting to membrane" pathway is frequently enriched in all three datasets. This suggests that the observed symptoms might be caused by a systemic change that affects both blood and the brain.

To further investigate the relationship between changes in the gene-expression in both tissues, we compared them to an additional bulk RNA-

sequencing dataset from the ROSMAP cohort (Synapse https://www.synapse.org/#!Synapse:syn3388564, Supplementary Fig. 8e). Although fold-changes in the brain single-cell and bulk data showed a significant positive correlation (overall correlation: 0.14, p-value: 0.00025), we did not find a significant correlation of the PBMC single-cell with the brain bulk data (Supplementary Fig. 8f).

### Access for the community
To allow easy, fast and uncomplicated data access to our large-scale findings, we developed a webserver for the dataset. The web-resource allows the visualization of gene-expression values (Supplementary Fig. 9a, b). The direct access to the list of de-regulated genes and its visualization (Supplementary Fig. 9c) allow fast hypothesis-testing.

In addition to the scRNA-seq data used for the analysis shown in this manuscript, the webserver also allows researchers to access the longitudinal data, brain volume measurements and CSF-marker values.

Our dataset contains 32 patients with more than 1 time-point (Supplementary Fig. 10a) and 3 with more than 2. All patient with 3 Visits were female and we observed differences in the age-distribution between male and female patients in this subgroup (Supplementary Fig. 10b). The average time between the first and second visit was less than a year for AD and between 2 and 3 years for the other diseases

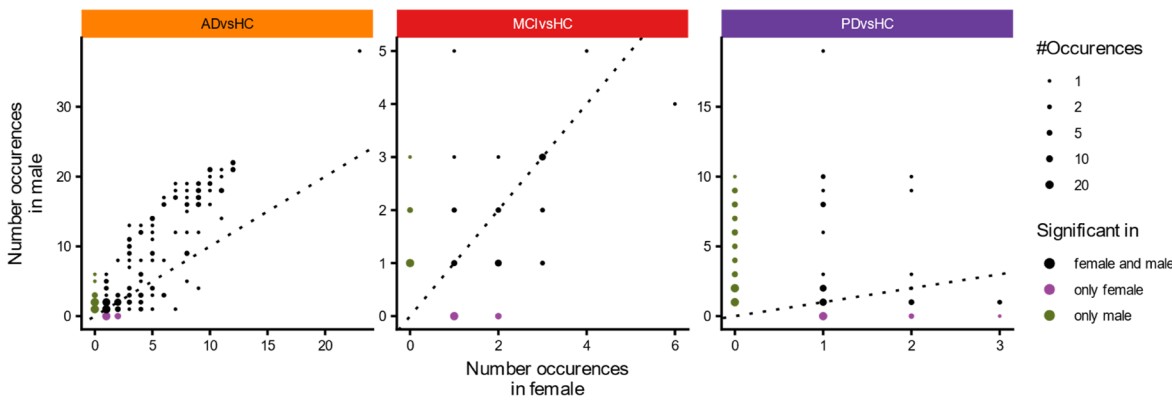

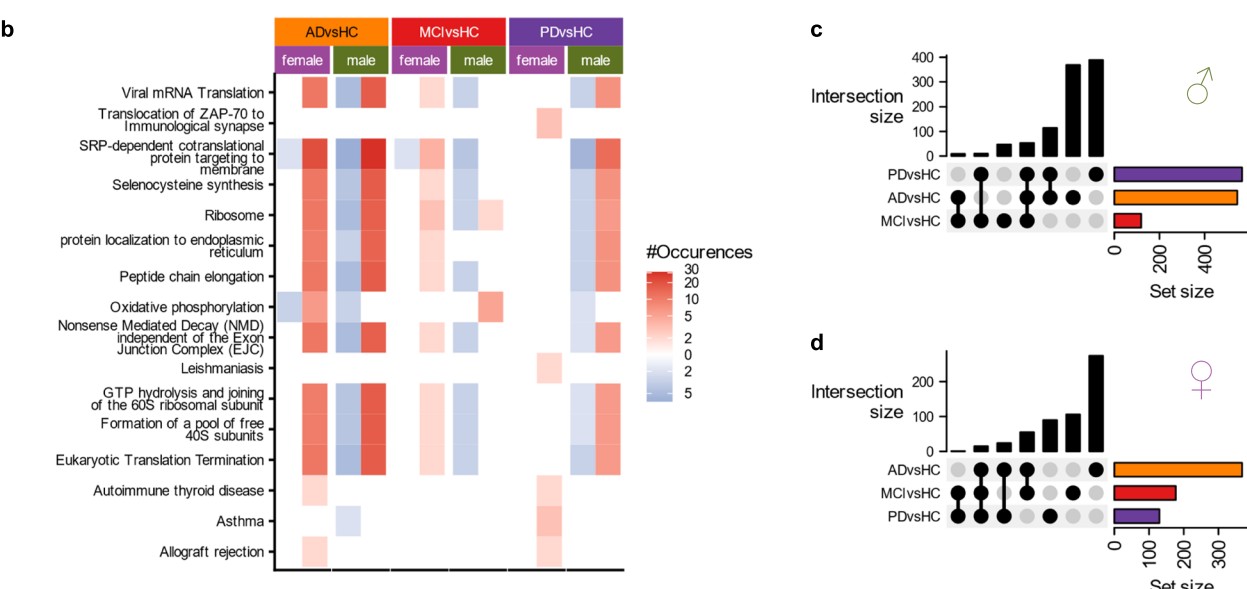

**Fig. 4 | Pathways analysis shows sex-dependent changes in PD and MCI. a** The number of cell types in which a pathway is enriched or depleted in males and females in the different comparisons. The dots are colored if they occur only in males or in females. **b** Number of occurrences of the top 5 most frequently enriched and depleted pathways in each comparison and sex. **c, d** Size of the overlap of the pathways in the different comparisons, independent of cell type for males (**c**) and females (**d**).

(Supplementary Fig. 10c). When correlating the cell-type proportions with the number of the visit, we observed no significant correlations (Supplementary Fig. 10d). Generally, both the cell-type proportions of the same patient at different visits and between different patients show large fluctuations (Supplementary Fig. 10e).

Although the statistical power of the longitudinal data is limited, we included it on the website, allowing interested researchers to study this data in more detail. We additionally included the data of the brain volume measurements (Supplementary Fig. 11a) and CSF-marker values (Supplementary Fig. 11b) for direct comparison. Similarly to the time-series data, we correlated it with the cell-type proportions (Supplementary Fig. 11c). We found significant correlations (adj. P-value < 0.05) mainly in healthy controls. We still include this data as a resource for other researchers.

Our dataset showed sex-dependent differences in the changes of the cell type composition and in the gene-expression profiles of neurodegenerative diseases and Parkinson's disease specifically.

## Discussion

The era of single-cell -omics has enabled studying human diseases at an unprecedented cellular resolution combined with massive throughput, generating millions of transcriptomic profiles at ease. While large-scale single-cell analysis of fresh-frozen post-mortem brains seems to come in close reach for the upcoming years, it is of great importance to understand which systemic changes precede and co-occur neurodegeneration in the human body as to inform our current understanding of early biomarkers as well as individualized risk or progression markers. Our dataset is composed of 290 gene-expression profiles of PBMCs from MCI, AD, PD, PD-MCI and HC individuals and includes additional information about Brain volume measurements and CSF-biomarker levels of known AD biomarkers. This dataset allows us to generally study disease- and sex-specific changes in patients with neuro-degenerative diseases.

To study the exact mechanisms, a more specialized dataset with more samples (e.g. for AD)[48] and additional demographic and clinical information (e.g. medication use, other disease, lifestyle factors,...) would be necessary. Changes in the sub-cell types and potential sub-groups in the cell clusters furthermore need to be studied in more targeted experiments, as the number of cells in each of these clusters is limited.

When comparing the cell type composition, we found large differences between the changes in men and women that are often specific for certain sub-cell types. Changes in B cells and Monocytes in

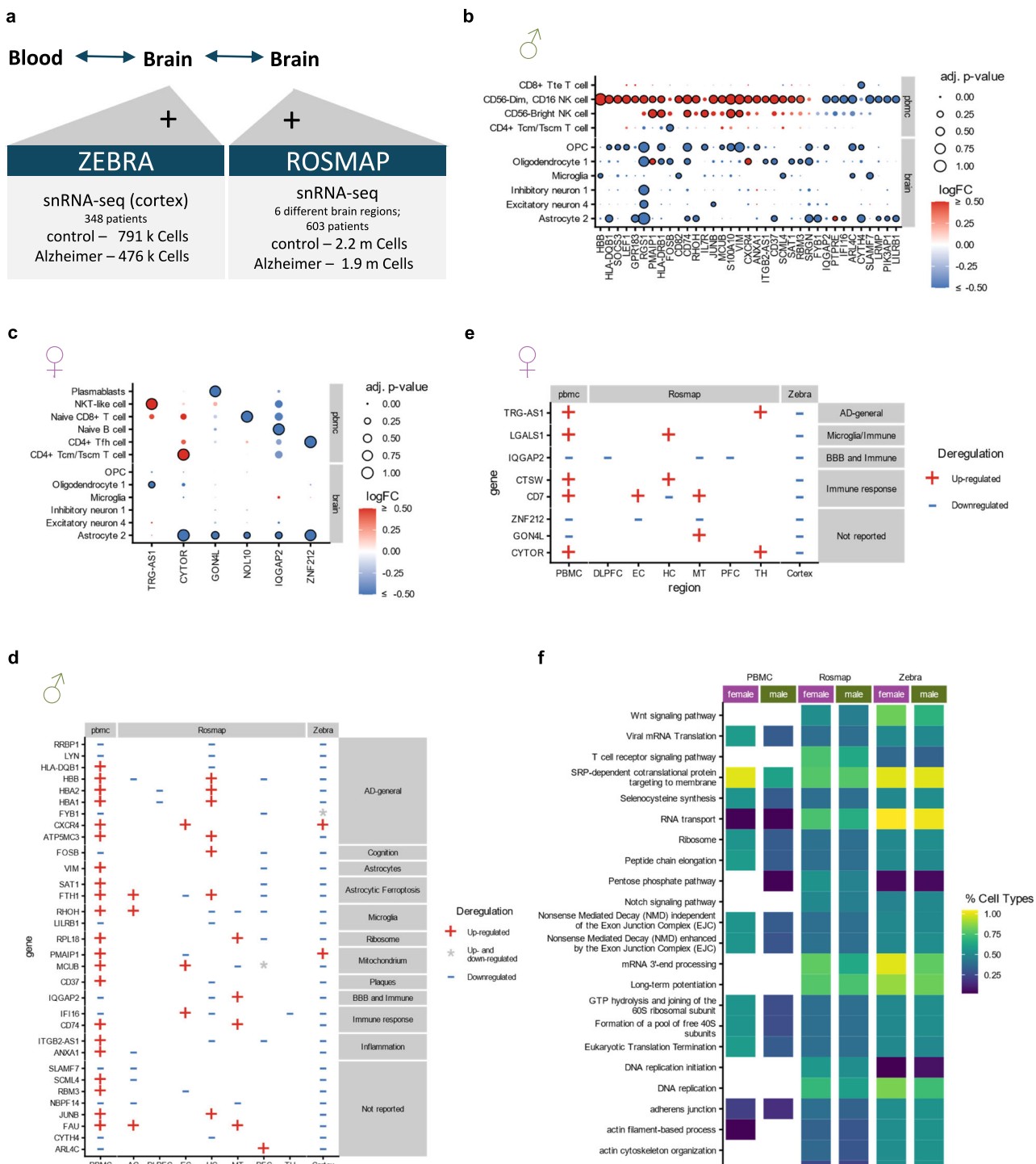

**Fig. 5 | Comparison of gene-expression in brain and blood of Alzheimer's patients. a** Single-cell RNA-sequencing data from the cortex from the ZEBRA-dataset was used for the comparison of changes in PBMC samples and brain samples. **b** Fold-changes in PBMCs and cortex of the genes with an adj. p-value < 0.05 (see Methods) in the PBMCs the brain in males. **c** De-regulation of genes in PBMCs and cortex with an adj. p-value < 0.05 in PBMCs and the brain and an abs. log2 fold-change bigger than 0.6 in the PBMCs (see Methods). **d, e** Deregulation of the significantly de-regulated genes in all three datasets (see Methods) and the direction of de-regulation in males (**d**) and females (**e**) ( + : Up-regulated in all cell-types, -: Down-regulated in all cell-types and *: Mixed signals). Genes are annotated with the context in which they have been previously reported in for Alzheimer's (see Supplementary Table 3). **f** Top 10 most frequently enriched or depleted pathways in both brain datasets and in PBMCs and the proportion of cell-types they are enriched/depleted in.

Alzheimer's disease were inconsistent across male and female patients. It has been previously shown that depletion of B cells from mice led to cognitive deficits and an increase of Aβ plaques[58]. B cells also play an important role in the regulation of local immune responses and might thus indicate inflammatory states in Alzheimer's. The increased levels of monocytes in AD can furthermore be a sign of chronic inflammation and may have a negative effect on structures like the blood brain barrier (BBB)[59]. In contrast to the large sex-specific differences in the PD cell type compositions that we observed, previous studies mainly found changes in the T-cell population of Parkinson's patients. Of note,

the findings of the previous studies sometimes contradict each other. This can be partially explained by the large fluctuations in the cell type composition that can be observed even of the same patient[60] so that differences in the cell type composition can vary largely between samples. Moreover, some of our findings have not yet been documented, or only partially align with previous studies. This could be attributed to biases in the cohorts used (such as differences in ethnicity, disease severity, or other factors), technological limitations, or other biases.

To identify sex- and disease-specific gene signatures, a machine-learning model was applied to the dataset (Supplementary Notes 1, Supplementary Data 8). Testing out a random forest-based feature selection and three other machine-learning methods, we concluded that the application on an Alzheimer's disease dataset is too complex and would require a larger dataset (ideally with different studies) to reliably predict disease-specific features across patients.

The data furthermore suggest sex-specific differences between gene-expression signatures of PD, more similarities, but still sex-specific differences in AD. Sex-specific differences in PD have been previously found in Monocytes[27] but sex-independent expression-changes were reported in T-Memory cells[61]. We have found an anti-correlation of gene-expression changes in male and female indicating large differences between the disease-specific changes in PD, while the sex-specific differences in AD can be mainly observed by the effect size of the gene-expression change. We have seen that we found much more DEGs in female patients than in male patients. Even though this effect can be observed across all diseases in this study, it is particularly strong in AD, which has previously been reported by Mathys et al. [11]. In Parkinson this effect has been described previously in blood Monocytes[27], whereas the exact opposite case has been observed in the brain[62].

The changes that were observed in Alzheimer's disease independent of the sex mainly focused on genes from the "SRP-dependent cotranslational protein targeting to membrane" pathway. This pathway has been previously reported in Alzheimer's disease[63] and Parkinson's disease[64,65]. This indicates similar changes in male and female Alzheimer's patients and male Parkinson's patients. Changes in those genes have been showing an increased expression during microglia activation, a state that has previously been associated with Alzheimer's[63].

The "SRP-dependent cotranslational protein targeting to membrane" pathway has also been observed in brain samples from both the ZEBRA and the ROSMAP dataset. The genes involved thus do not only show changes in PBMC, but also the Brain. The genes of this pathway have previously been discussed in the context of periodontitis, a condition mainly caused by *Porphyromonas gingivali*[63,66]. This bacterium is known to secrete gingipains, which have been linked to Alzheimer's pathology in mice and humans[67,68].

Diving deeper into the changes that are observed in both the blood and the brain, we found 30 overlapping genes in males and 7 in females that are significantly deregulated in both blood and the brain. A majority of those genes have been previously reported in relation to Alzheimer's disease in different tissues and cell-types (Supplementary Table 3). These include genes related to changes in the Immune system and the Blood-brain barrier (especially in females) and changes related to Astrocytes and Microglia. This indicates general changes in the immune system, possibly in relation to an infection (such as herpes simplex) or age-related inflammation processes. The infection with Herpes simplex can lead to a higher risk of developing Alzheimer's in APOE4 carriers[52,53] and is discussed to play a major role in the disease development.

As patients of the ADRC cohort are continuously monitored, two major shortcomings of our study will be addressed in the future. First, longitudinal trajectories will be important to trace down the temporal dynamics on the molecular level. Second, a patient-matched analysis of both brain and blood tissue will enable us to validate the here reported effects ruling out inter-individual variation.

In sum, our dataset indicates that there are strong sex-related differences in the neuro-degenerative diseases discussed in this study. While the exact relationship between the immune-system and the brain remains unknown, this study highlights the importance to consider possible influences of the immune system and the sex on the development of these diseases. Beyond the analyses that we provide in the present study, we are convinced that exploring the very broad dataset supports a broad range of further research topics. We thus make not only the raw data available but also implemented a convenient web interface with substantial functionality to explore the data (https://www.ccb.uni-saarland.de/adrcsc).

## Methods

### Cohort

The patients included in the study were enrolled by the Stanford Alzheimer's Disease research Center (ADRC). The group of patients includes patients diagnosed with mild cognitive impairment (individual patients n = 51, total samples from the patients $n_t$=63; MCI), Alzheimer's disease (n = 31, $n_t$=43; AD), Parkinson's disease (n = 55, $n_t$=61; PD), PD with MCI (n = 20, $n_t$=24, PD-MCI), and healthy individuals (n = 154, $n_t$=172, HC). We selected patients with respect to a similar ratio of both biological sex and age between the subgroups. The Stanford University IRB approved the study and all patients provided written informed consent. Due to the strict quality control, the final dataset only contains a total of 290 samples from 257 individuals. CSF measurements were performed for NFL, UCHL1, Tau and GFAP (Quanterix4plex), Ab40, Ab42 and Tau (Quanterix3plex) and pTau181 (Quanterix). Brain Volume measurements were performed using PET and MRI imaging.

### Single-cell suspension preparation

Frozen PBMCs vials were rapidly thawed in a 37 °C water bath for 2 minutes, and the vials were removed when a tiny ice crystal was left. Thawed PBMCs were quenched with 13 ml 37 °C pre-warmed 1X phosphate-buffered saline (PBS, Thermo Fisher Scientific, #10010031) supplemented with 10% fetal bovine serum (FBS, Thermo Fisher Scientific, #A3160601). Cells were centrifuged at 300 x g for 10 minutes at room temperature. The supernatant was removed, and cell pellet was resuspended in 3 ml 1X PBS containing 10% FBS, passed through a 40 μm cell strainer (Falcon, #352340), then centrifuged at 300 x g for 10 minutes at room temperature. Dead cells were removed by magnetic beads purification (Miltenyi Biotech, #130-090-101) according to the manufacturer's protocol. Cells were resuspended with cell resuspension buffer at a concentration of 1000 viable cells/μl.

### Single-cell RNA-seq with DNBelab C4 system

The DNBelab C Series Single-Cell Library Prep Set (MGI, #1000021082) was utilized as previously described[49]. In brief, single-cell suspensions were used for droplet generation, emulsion breakage, beads collection, reverse transcription, and cDNA amplification to generate barcoded libraries. Indexed scRNA-seq libraries were constructed according to the manufacturer's protocol. The DNA nanoballs (DNBs)-based libraries were sequenced by the ultra-high-throughput DIPSEQ T1 sequencer at China National GeneBank (CNGB). The read structure was paired-end with Read 1, covering 30 bases inclusive of 10-bp cell barcode 1, 10-bp cell barcode 2 and 10-bp unique molecular identifier (UMI), and Read 2 containing 100 bases of transcript sequence, 10-bp sample index.

### Primary single-cell RNA-seq data processing

Raw sequencing reads from DIPSEQ T1 sequencer were filtered and demultiplexed using PISA (version 0.2) (https://github.com/shiquan/PISA). Reads were aligned to hg38 genome using STAR[69] (version 2.7.4a), and sorted by sambamba[70] (version 0.7.0). Genes were

annotated according to GENCODE[71]. Cell versus gene UMI count matrix was generated with PISA.

## Quality control and data processing

For each count matrix, SoupX[72] (1.4.8) was used to estimate and remove ambient RNA contamination. Only samples with at most 10% estimated contamination were kept for further analyses. Subsequently samples were filtered such that each sample contained at least 300 cells, each cell contained at most 7% mitochondrial gene counts and between 300 and 4000 genes were expressed per cell. Doublet filtering was also performed sample-wise using the R Bioconductor package scDblFinder[73] 1.2 with an expected doublet ratio of 1% per 1000 cells. Next, all samples were merged.

## Dimensionality-reduction and clustering

Dimensionality-Reduction and clustering were performed using Seurat (version 4.0.0)[74]. First the data were again normalized, scaled and the top 2000 variable genes were selected using Seurats NormalizeData, FindVariableFeatures and ScaleData functions. For the diemensionality-reduction, we used the RunPCA and the RunUMAP function with the first 10 dimensions, which was determined using Seurats ElbowPlot-Function. We performed clustering using the FindNeighbors-function with the first 10 dimensions and the FindClusters-Function with a resolution of 0.1.

## Cell type annotation

This resulted in multiple clusters which could be assigned to B cells, T and NK cells, myeloid cells, Plasma cells and a cluster with mixed cell populations. To identify cell subclusters we next performed a second round of dimensionality-reduction and clustering of each of the previous clusters. The T/NK cell clusters are first processed independently for each batch for the filtering and then merged afterwards to perform cell type annotation. We then filtered out sub-clusters with more than 50% cells that originate only from one sample and repeated the dimensionality-reduction and clustering on the filtered data, as described for the complete set of cells. We repeated this step until no cells are filtered out. Cell sub-clusters are then checked for known marker-genes. Clusters that did not express known markers of PBMC cells or that are indicated to be low-quality or doublets are removed from the dataset, followed by another dimensionality-reduction and clustering step. The T/NK cells of the different batches are then merged, dimensionality-reduction and clustering were performed and the cells were split into CD4 + , CD8+ and NK cells before continuing with the sub-clustering and the cell type annotation. After having filtered out the unwanted cells, the final sub-clusters were annotated based on known markers from literature.

## Low dimensional embedding and density computation

An overall cell density per cell type and patient cohort was determined by first estimating the cell density per sample and cell type with the two-dimensional kernel density estimation implemented in the kde2d function of the MASS R-package (version 7.3-55). Subsequently, the density estimates were aggregated per patient cohort by averaging the obtained density estimates.

## Cell-type composition analysis

Changes in the cell-type composition were evaluated using an unpaired Student's t-Test on the raw cell-type proportions and the p-values were corrected using the Benjamini-Hochberg procedure. scCODA (version 0.1.9)[75] was first used with an automatically selected reference cell type and Age and AgoE as covariates. Using the selected reference cell type (CD4 + T cell for the broad annotation and Treg CD4+ cell for the fine annotation), the analysis was repeated separately for males and females. The corrected Fold-changes were reported, and significant changes were determined using a p-value cut-off of 0.05.

## Differential gene expression analysis

Differentially expressed genes on the PBMC data were determined with the pseudobulk approach implemented by the muscat[76] package (version 1.6.0). Cells were aggregated with the aggregate data function at cell type and sample level by summing up the SoupX corrected counts. The pbDS function was then used with the limma-voom[77] package to calculate the set of differentially expressed genes. The design matrix included besides the diagnosis, sample batch and sex as factors. Multiple testing adjustment was performed with the Benjamini-Hochberg procedure. Genes were subsequently filtered to be expressed in at least 5% of all cells of a cell type in at least one of the comparison patient cohorts. Genes were considered significantly de-regulated when they showed an absolute log2 fold-change above 0.5 (i.e. a log2 FC < -0.5 or > 0.5) and an adjusted p-value smaller than 0.05. Adjusted p-values were reported as given by the muscat package.

## ZEBRA Brain Atlas preparation and DEG analysis

The study collection, pre-processing, and aggregating of the brain single-nucleus data was performed as described in Flotho et al.[51]. The Blanchard et al.[78] was filtered and the Hardwick et al.[79] dataset was removed due to overlaps with other studies in the dataset. The marker genes have been computed using a pseudo-bulk approach with edgeR (v3.36.0). Only cells from studies which reported a particular gene were considered for computing the DEGs. If the cells for computing the DEGs originated from more than a single study the study ID was used as a latent variable in the edgeR design matrix. For significance testing the glmQLFit method was used. Finally, the p-values have been adjusted using Benjamini-Hochberg correction of p.adjust method from the R stats package (v 4.1.3).

## ROSMAP Brain Atlas preparation and DEG analysis

We collected and harmonized a scRNA-seq AD/CT dataset from state of the art studies using sample from the Religous Order and Aging Project (ROSMAP)[80]. In particular we used the studies syn52293433, syn18681734, syn51062116 and syn2580853.

We utilized the Synapse client for Python (synapseclient version 4.5.1) with Python 3.11.10 to acquire all necessary metadata. For realigning the counts we use the CellRanger v8.0.1 software with the 2T2 reference genome. We removed redundancies in the data by overlapping cells and shared donors across studies and removed all duplicated cells from our downstream analysis. We apply the following QC thresholds: Genes have to be present in at least 3 cells, each cell with less than 100 unique genes has been discarded. We remove putative doublets using scrublet v0.2.2[81] with default parameters, and run CellBender v0.3.2[82] as well with defaults.

We use the Allen Brain Institutes MapMyCells (RRID:SCR_024672) for cell type labeling, here only cell types with more than 100 cells have been considered for further analysis. For computing DEGs we followed the already described methodology (Methods: Differential gene expression analysis) and included all NCI and AD cases in our comparison.

## Pathway enrichment analysis

Enriched pathways on the whole gene-set were determined by performing an unweighted gene set enrichment analysis using GeneTrail 3[83] and the "GO - Biological Process", "GO - Cellular Component", "GO - Molecular Function", "KEGG - Pathways", "Reactome - Pathways", "WikiPathways" and "Pfam - Protein families" databases. Genes were ordered by the -log10 adjusted p-value and the absolute log2 fold-change. Only genes which were used by the models were considered. Pathway analysis for pre-selected gene subsets were performed using over-representation analysis using GeneTrail 3 with the same database[83]. Pathways with an adjusted p-value smaller than 0.05 were considered significantly enriched or depleted. P-values were reported as given by GeneTrail.

## Cell-cell communication analysis

Changes in the cell-cell communication were evaluated using the CellChat package[54] (version 1.5.0).

## Literature Search

We performed a manually curated literature-research using PubMed-Search by combining the key-words "Alzheimer's" and "Parkinson's" with "blood", "peripheral immunity", "peripheral blood", "pbmc" or " CSF marker" or "brain" with and without the keyword "sex" and selected papers that provide further information and previous findings on the topics of this paper. For the single-cell and bulk PBMC datasets of Alzheimer's and Parkinson's, we performed a manually curated literature-research using PubMed-Search with the terms "transcriptomics", "single-cell", "pbmc", "human", "RNA-seq", "blood", "parkinson", "alzheimer" and "neurodegeneration" and selected all single-cell and bulk RNA-seq datasets.

## Correlation analysis

All correlation values were determined using the Pearson's correlation coefficient (cor.test Function). P-values were adjusted using Benjamini-Hochberg. Correlations were considered significant if they had an adjusted p-value smaller than 0.05.

## Cosine similarity

The similarity between changes in the gene expression was calculated as the cosine similarity between the unfiltered gene-lists ordered by their fold-changes.

## Statistics and reproducibility

All test statistics used were conducted two-tailed, if not indicated otherwise. Computational tools requiring seed generators were set to use random but fixed initialization constants. Unless stated differently, p-values were corrected for multiple hypothesis testing using Benjamini-Hochberg. Unless stated otherwise, all analyses were performed using the baseline sample for each patient, with specifically defined analyses involving also longitudinal samples.

## Webserver implementation

The interactive visualization website was implemented using Django v2.2.16 for the backend and bootstrap v5.0.1 for the front end. For the visualization, plotly js v2.0.0 was used for the graphs and datatables v1.10.25 for the differential expression table. The data was stored in the h5ad format and accessed with anndata v0.7.6.

## Reporting summary

Further information on research design is available in the Nature Portfolio Reporting Summary linked to this article.

## Data availability

Raw sequencing data is freely available from the sequence read archive (SRA) using accession ID SRP312418. An interactive webserver to explore the scRNA-seq count data is freely available at https://www.ccb.uni-saarland.de/adrcsc. As metadata, we publish sex and diagnosis to comply to Nature's scientific data policy. Source data are provided with this paper.

## Code availability

The code used to analyze the data is available from GitHub[84]. Source data are provided with this paper.

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

## Acknowledgements

The study has been supported by the Michael J Fox Foundation under grant agreements 14446, 17047 (subc. 125491594) and MJFF-021418. The Schaller-Nikolich-foundation provided further support for the study. Computational resources used within this study were financed through the DFG under project 466168626. The development of the software pipeline has been supported by funds of the EU (Project 101057548-EPIVINF). We thank Shao Ning Pei, I-Jane Chen, Jody Beecher, Yongwei Zhang from Complete Genomics and Zhouchun Shang from MGI-SZ for their support.

## Author contributions

F.G., T.F., F.K., M.F., and T.W.W. performed computational analysis. F.G., T.F., and F.K. analyzed the data and assembled the figures. J.S.G. performed the literature research. F.G., T.F., F.K., and A.K. wrote the manuscript with input from all authors. T.F. and Q.S. performed primary data processing. E.Mass and D.M.G. assisted with celltype annotation. D.M.G., O.L., D.C. provided and organized samples. P.H. developed the webserver. E.N.W. performed the CSF biomarker measurement experiments and K.I.A. supervised this part of the study. E.Meese, D.M.G. and K.P. provided field-expertise and guided clinical interpretation. C.L., Y.L., C.C., Y.Y., J.X., M.J., Z.W., T.W. performed the sequencing experiments. L.L. and Y.H. supervised the sequencing experiments. A.K. and T.W-C. designed and supervised the study.

## Funding

## Competing interests

A.K. was advisor of the company Firalis, researching RNA-based biomarkers for AD, while preparing the manuscript. C.L., Q.S., Y.L., C.C., Y.Y., J.X., M.J., Z.W., T.W., L.L., and Y.H. work for the company MGI group while working on the project. The remaining authors declare no competing interests.
