## [Transparent Peer Review file · Nature Communications]

A single-cell atlas to map sex-specific gene-expression changes in blood upon neurodegeneration

Corresponding Author: Professor Andreas Keller

Version 0:

Reviewer comments:

Reviewer #1

(Remarks to the Author)

This is an excellent work by Grandke et al. applying machine learning algorithms to identify disease-associated alterations in blood cell type composition in a sex-specific manner, predominantly for B and NK cells. The work presents PBMC signatures in neurodegeneration across multiple disorders, combining expression-profiles of PBMC samples with snRNA-seq brain, levels of known AD-biomarkers in the CSF, and brain volume measurements. The results make a strong contribution to the current scientific debate and present the data interactively through a freely available website. The work is well designed, the methodology is sound and the manuscript is well written and suitable for publication after minor improvements (see below).

Content Points :

- 1) The authors reference a dataset of 290 samples from 257 subjects (L395). Therefore, they poses time-series follow-ups for a subset of patients. How was this data integrated in the main analysis? What was the number of timepoints available and their correlation to other metadata collected? Do the results different at a patient level between timepoints, and how?
- 2) Was the model Random Forest Classifiers also validated with external available data, such as it was referenced in Figure 1. If yes, what were the results?
- 3) The authors filtered the differentially expressed genes by an absolute log2fold-change above 0.5 and a adjusted p-value smaller than 0.05 (L501). Dysregulation is a phenomenon that includes genes that are up- and down- regulated. The mentioned filtering parameters correspond to genes that are significantly up-regulated (above 0.5). What about down-regulated genes (below -0.5)?

Structural Points :

- 1) Figure 1 as it is now could be considered / moved as Extended Data / Supplemental figure.
- 2) Figure 2 is referenced before Figure 1 (L109). The order of the figures should correspond to the order in which they are referenced in the manuscript.
- 3) Fig A1, A2 do not have an embedded legend.
- 4) Fig A4 b. Incomplete text description
- 5) Grammar and Typos:
 - a) (L47) additionally -> additional
 - b) (L314) "suggest a large ..."
 - c) (L511) (BH) unique abbreviation
- 6) There should be a uniform nomenclature.
 - a) (L111) "Figure" and "Fig" both used.
 - b) deregulation, de-regulation and dysregulation were used interchangeably.

(Remarks on code availability)

Reviewer #2

(Remarks to the Author)

This paper utilized a uniquely large, rich dataset (both in # of cells and # of samples) to investigate the relationship between PBMCs and neurodegenerative disease. The authors found that there were sex-specific changes in cell type proportion in a number of different diseases. Males with AD had higher levels of CD8+ T cells and females with MCI had higher levels of NKT-like cells and monocytes. They then built a random forest framework to try and identify a small number of genes that could be used to distinguish between cells from healthy donors vs. patients with disease. They found that they were able to build 80% accurate models on cells and up to 90% accuracy on patients. They then conducted sex-stratified differential gene expression analyses between healthy controls and cases with neurodegenerative diseases. They found that the number of DEGs varied greatly between sexes across the same cell type, especially in AD. Through further analysis, they determined that the level of shared difference between diseases is common across sexes but that the signatures vary greatly between sexes. Finally, they wanted to compare the signal in PBMCs to the signal in the brain, so they took the dysregulated genes in PBMCs and looked for expression changes in cells from the brain. They found no correlation between gene expression in females and a slight anti-correlation in males. Overall, the authors seek to promote the utility of their large dataset and prompt others to consider the impact of sex when determining biomarkers of disease. The major strength of this work is the dataset generated, which will be a tremendous resource to the community. Several methodological weaknesses reduce enthusiasm regarding the main scientific analyses conducted and, thus, on the conclusions made from these analyses. Specifically, the random forest analysis is fraught with issues, noted in detail below, and does not seem to add much to the manuscript. There is also a missed opportunity to highlight the rich biomarker data that is available on their web tool (which is not really included in any of the analyses). Enriching the presented analyses and potentially expanding them to further explore correlations between cell types or gene expression could also add to the scientific value of this exciting dataset.

Main Concerns:

1. Analyses are being run on raw cell proportions. It is prudent that the authors also report results from adjusted analyses of cell proportions, for example, using a regression model with adjustments for covariates (i.e., age, APOE e4 status (especially since it is sex split)), which is a common practice for the field and found in other, similar studies.

2. The rationale for the random forest analysis is unclear. Among other issues, the fact that most of the genes identified are not differentially expressed raises some major red flags as to the validity of the method. In particular:

a. It is not clear what the objective of the model is.

i. If the goal is to make predictions on disease status...

-I would love to see a comparison between these models and models using DEGs from the authors' other analyses (I think DEGs would do as well or better and make more sense scientifically as a starting point)

-The authors could potentially use the same dropout method but use different DEG thresholds as a starting point

-Most other models trying to do this are complex deep learning models with full data, not 2000 genes; what is the justification for using this subset? Is there precedent for this being a valid approach?

ii. If the goal is to identify genes that are different...

-I think differential expression analysis is a much better way to do this; if this is not the case, then a very clear explanation and justification for this should be presented in the manuscript

b. There are numerous missing methodological details, which diminishes transparency & reproducibility

i. There is no mention of how feature importance is assessed. I assume it is a common python or R package, but it would be good to know which one as this is an important detail.

ii. I did not see any mention of hyperparameters for the random forest model (i.e., max depth, number of trees).

iii. Most ML methods use AUC or F1 as a metric rather than accuracy. If the authors choose not to use either of those, then a clear justification for why accuracy is a reasonable metric to use for evaluating the findings is required.

3. The web server that they have set up looks like a great tool for exploring the data and provides a good community resource. The title of the paper also seems like it is designed to promote this feature. However, the results generated by this tool are not shown in the main body of the paper. I think that to show the utility of the tool and the additional biomarker and longitudinal data it houses, an analysis using the data highlighted within the tool should also be included in the manuscript.

Minor Concerns:

1. I would like to see a true Table 1 with a demographic breakdown rather than a series of graphs as provided in Figure 2. It is very difficult to derive meaningful information from the data as presented in Figure 2. It would also be more informative to provide statistical comparisons between each of the covariates between the sexes since this is one of the main contrasts presented in the manuscript.

2. The paper only shows sex-stratified results for cell proportion. It would also be helpful to provide a non-sex-stratified version so that it is possible to see if the signal is maintained but only statistically significant in smaller n populations (e.g., this was done for DEGs and there were not many overlapping hits).

3. The Venn diagrams in Figure 2 are not very easy to decipher and many of the labels are hard to read. I would recommend the authors explore alternative strategies for presenting this information in a clearer way.

4. When pathway analysis for DEGs is performed, only AD and PD-specific KEGG was run. It would be helpful to also include more general GO analysis, especially since the authors don't find any significant hits and openly admit that the pathways might be biased against sex-specific signals.

Other:

- Most of the findings from the paper have to do with sex differences, however, the title makes no mention of this; modifying the title to highlight the main research findings would help readers interested in this topic find this resource
- Figure 1: it looks like some of the data points are duplicated (Wang x 2? Sirkis x 2?) and/or have incorrect values (e.g., does Sirkis really have only 100s of cells?).
- The references for Figure 1 data are incomplete
- Figure 3c: the dots and the stars look very similar given the small size. Is there an alternative way to show this data?
- Extended data 7c-d: Y axis labels missing from plots

(Remarks on code availability)

Reviewer #3

(Remarks to the Author)

The topic of explore the peripheral immune system's role in neurodegenerative diseases is intriguing. While the results hold potential merit, the presence of several serious conceptual and technical flaws in methods and a lack of adequate validation on external datasets significantly reduce our support for publication in its current form.

The following are some considerations that could further improve the manuscript:

Major:

1. The demographic and clinical information of the sample population is not provided. The study should consider and discuss potential confounding factors such as age, medication use, other disease, lifestyle factors, etc., that might influence PBMC gene expression.
2. The study identifies changes in gene expression and cell type abundances, but the functional implications of these changes are not discussed. Further investigation and discussion about how these molecular changes translate to clinical symptoms or disease mechanisms would be valuable.
3. The manuscript does not provide a comparison of the random forest method with other machine learning techniques. It is important to ensure that this method is indeed the most suitable one for the specific data and objectives of the study.
4. In the method section, the author should explain how the hyperparameters in random forest were tuned and provide evidence that the selected parameters are indeed optimal.
5. It is difficult to understand why most of the features selected by the machine learning model are not differentially expressed. The authors should provide a more detailed explanation of the possible biological mechanisms underlying this observation.
6. The p-value adjusted method is not used in the correlation analysis. Given the large number of cells analyzed, adjusted p-values should be reported for the correlation analysis.
7. In the manuscript, the authors acknowledge several previously generated single-cell datasets. They mention some consistencies and inconsistencies between their findings and those of other studies. However, the comparison can be improved by integrating in-house and public data. The incorporation of other datasets may increase statistical power and provide more insights from independent cohorts.
8. It is unclear how immune cells in PBMCs influence or are associated with disease initiation and development. Although the authors identified a few genes that are differentially expressed in both PBMCs and the brain, they did not demonstrate the function of these genes in a cell-type-specific manner, as the same genes may have different functions in different cell types. Additionally, how PBMCs are linked to brain changes is unclear.
9. The machine learning model designed to distinguish between healthy and diseased individuals needs further clarification and potential strengthening in the following aspects:
 - i. The authors need to clarify that the accuracy was calculated using cross-validation and not independent testing. Independent validations will better assess the model's accuracy.
 - ii. Does the abundance of cell types in PBMCs contribute to the prediction? Since the authors observed differences between healthy donors and diseased individuals, it is reasonable to hypothesize that cell abundance may have some predictive power. Will integrating cell abundance with gene signatures improve the prediction power?
 - iii. The best accuracy on the cell level was achieved with a model trained on B cells, and on the patient level, the highest accuracy was achieved using male NK cells. What are the functions of the signature genes identified by the machine learning model? How do these functions influence disease initiation and development?
10. Differentially Expressed Genes (DEG) Analysis:

What is the rationale behind the DEG analysis within cell types? Does the same cell type express different genes in different patients? As the authors already annotated cells at a high resolution, do they expect subpopulations within the current

annotations? What are the differentially expressed genes within the same cell type, and what are their functions and impacts on disease development? This part needs further clarification.

11. PBMC and Brain Disease Signatures:

In the section "PBMC disease signatures are inversely correlated with changes in neural cells," the authors integrate data generated using both brain and PBMCs and aim to show how changes in PBMCs are associated with changes in the brain in disease. Several questions need to be addressed in this section:

- i. Are there any shared cell types between PBMCs and the brain in patients?
- ii. How do changes in PBMCs influence changes in the brain or vice versa?
- iii. What are the genes (43 genes in males and 8 genes in females) that are deregulated in both PBMCs and the brain, and what are their functions?
- iv. The mechanism by which the brain interacts with PBMCs in patients remains unclear and needs further elucidation.

(Remarks on code availability)

The code repository includes scripts for generating all the figures in the manuscript as well as most of the code used for data processing.

Version 1:

Reviewer comments:

Reviewer #1

(Remarks to the Author)

I would like to thank the authors for this in depth revisions of the manuscript. My concerns have been adequately addressed! I see a very valuable and sound manuscript that merits it consideration for this journal.

Reviewer #2

(Remarks to the Author)

The authors addressed all of the main feedback from the initial review and replaced the most problematic analysis with a new, more compelling one. The authors' updates to the statistical methods are more robust and this is greatly appreciated. However, there remain some minor issues that would be ideal to address:

1) While the authors created a Table 1 and did some age comparisons between sex in each disease state, the significance from this test should also be included in the table. For full transparency, it would be helpful to provide a comparison across all covariates between sexes in the table as well (including statistical comparison results).

2) While it was very helpful that the authors added the years to the figure showing other PBMC datasets and included a separate reference table, one of the datapoints still shows only 100 cells (which is incorrect, as noted by the authors themselves in the response to reviewers document and throws off the scale of the graph. Please ensure all values are accurately updated in this figure.

Once these minor issues are addressed, the manuscript will represent a compelling body of work worthy of publication.

Reviewer #3

(Remarks to the Author)

The authors have made some adjustments in response to our comments. However, we think most of our concerns remain unaddressed.

Demographic and Clinical Information: While the authors acknowledged the lack of detailed clinical data, such as medication use and lifestyle factors, their response is unsatisfactory. Merely controlling for age and sex without considering other potential confounding factors significantly undermines the validity of their conclusions. The absence of these important details should not be dismissed, and this aspect was not addressed with the rigor required.

Functional Implications of Changes: The added pathway analysis is a step forward, but it feels more like an attempt to patch up the original concerns than a fundamental solution. The explanation remains superficial, and the authors fail to adequately explore the broader clinical or biological implications of the observed changes. More in-depth exploration of how these findings relate to disease mechanisms is still lacking.

Random Forest Analysis: Simply removing the Random Forest model is an evasive move. Instead of answering legitimate questions about their methodology, they have sidestepped the issue altogether. This approach does not address the core problem of methodological robustness. The absence of any comparison to other machine learning models or justification for their initial approach leaves a significant gap in the study's reliability.

PBMC and Brain Disease Signatures: The additional pathway analyses, while informative, do not resolve the deeper concern about the connection between PBMC and brain data. The authors fail to address how the changes in PBMCs might relate to brain disease mechanisms in a meaningful way. Their reliance on pathway overlaps does not provide the

necessary mechanistic insight, and their explanation remains overly general.

Hyperparameter Tuning for Random Forest: Although the Random Forest model has been removed, their explanation of hyperparameter tuning is irrelevant now. The authors should have demonstrated why their approach was valid in the first place rather than simply avoiding the issue.

Machine Learning Model Explanation: Again, the removal of the Random Forest analysis is an unsatisfactory way of dealing with this concern. Instead of improving the model or providing clear justification for their method, they have taken the easy way out. This undermines the overall rigor of their study.

In summary, we think the rebuttal falls short of adequately addressing the core concerns raised. Instead of improving their methodology or providing detailed explanations, the authors have opted to remove the problematic aspects of their analysis without properly defending or revising their approach. This is not an acceptable solution, and many of the fundamental questions remain unanswered. We, therefore, recommend not accepting this paper.

Version 2:

Reviewer comments:

Reviewer #2

(Remarks to the Author)

Overall, the main strength of this study remains the incredible data resource and exploration portal that have been generated, which will be of great interest to a broad audience of researchers. The findings presented highlight the importance of sex differences in biology and omics research and will hopefully seed future work that can further explore sex-differences using this and other data resources. The authors have largely addressed most of the prior comments and suggestions, and the methods and results presented are satisfactory. There are a few minor additional items that would be worth addressing to enhance the presentation of results and accuracy of the study prior to final publication, though addressing these would not preclude the manuscript's consideration for publication.

Minor comments:

- In the comparison of males and females in each disease by number of deregulated genes in Figure 3a, it might be easier to spot the difference if each disease was lined up next to the other.
- The section discussing the failure of the Random forest model would be better placed in the Discussion section rather than in its current location.
- Please ensure the number of patients in ROSMAP is accurately reported.

Reviewer #3

(Remarks to the Author)

We appreciate the authors' efforts in addressing our concerns. We believe the dataset has the potential to benefit the community, and the findings offer valuable insights for Alzheimer's research.

Version 3:

Reviewer comments:

Reviewer #4

(Remarks to the Author)

Answers to the reviewer comments

Reviewer #1 (Remarks to the Author):

This is an excellent work by Grandke et al. Applying machine learning algorithms to identify disease-associated alterations in blood cell type composition in a sex-specific manner, predominantly for B and NK cells. The work presents PBMC signatures in neurodegeneration across multiple disorders, combining expression-profiles of PBMC samples with snRNA-seq brain, levels of known AD-biomarkers in the CSF, and brain volume measurements. The results make a strong contribution to the current scientific debate and present the data interactively through a freely available website. The work is well designed, the methodology is sound and the manuscript is well written and suitable for publication after minor improvements (see below).

We thank the reviewer for the generally positive feedback. Below you find a detailed response to every comment.

Content Points:

- 1) The authors reference a dataset of 290 samples from 257 subjects (L395). Therefore, they pose time-series follow-ups for a subset of patients. How was this data integrated in the main analysis? What was the number of timepoints available and their correlation to other metadata collected? Do the results differ at a patient level between timepoints, and how?

We apologize being not clear in this aspect. Although the time-series data provides an interesting addition to the study, the somewhat limited number of samples lead us to leave out the time-aspect from the main analysis but add it to the supplemental material. Other aspects such as the pathway analysis were more central for the main analysis in our view. Nonetheless we think that even if the number of longitudinal samples is limited it might be of value for others, e.g. to validate findings in their studies. Thus, we decided to leave those samples in. We stated this explicitly in the updated manuscript in the Results: “The downstream analysis described in this paper was performed using the baseline sample of each patient unless stated otherwise.”

But to answer some of the questions posed here, we added an additional Supplementary Figure 10, going into more detail into the number of samples, the age, sex and diagnosis of the patients as well as the time between visits. We also allow interested users to study the data on our webserver.

- 2) Was the model Random Forest Classifiers also validated with external available data, such as it was referenced in Figure 1. If yes, what were the results?

We did originally not perform an external validation, mostly because compatible data sets at scale were lacking. This is not only in the view of this reviewer but also according to our opinion a challenge, because we can't prove the generalizability of the model. Due to this comment and the concerns of Reviewer #2 and #3, we decided to remove the Random Forest model from this manuscript. We instead added a stronger emphasis on pathways (see new Figure 4) and the webserver as suggested by Reviewers #2 and #3.

- 3) The authors filtered the differentially expressed genes by an absolute log₂fold-change above 0.5 and a adjusted p-value smaller than 0.05 (L501 -> new L ...) Dysregulation is a phenomenon that includes genes that are up- and down- regulated. The mentioned filtering parameters correspond to genes that are significantly up-regulated (above 0.5). What about down-regulated genes (below -0.5)?

We apologize for this miscommunication. The absolute log₂fold-change means that we take the absolute value of the log₂ fold-change to filter the genes. That means that if we select genes with an absolute log₂ fold-change above 0.5, we do select both genes with a log₂ fold-change smaller than -0.5 and genes with a value above 0.5. We clarify this again in the Methods section: "Genes were considered significantly de-regulated when they showed an absolute log₂ fold-change above 0.5 (i.e. a log₂ FC < -0.5 or > 0.5) and a adjusted p-value smaller than 0.05."

Structural Points :

- 1) Figure 1 as it is now could be considered / moved as Extended Data / Supplemental figure.
- 2) Figure 2 is referenced before Figure 1 (L109). The order of the figures should correspond to the order in which they are referenced in the manuscript.
- 3) Fig A1, A2 do not have an embedded legend.
- 4) Fig A4 b. Incomplete text description
- 5) Grammar and Typos:
 - a) (L47) additionally -> additional
 - b) (L314) "suggest a large ..."
 - c) (L511) (BH) unique abbreviation
- 6) There should be a uniform nomenclature.
 - a) (L111) "Figure" and "Fig" both used.
 - b) deregulation, de-regulation and dysregulation were used interchangeably.

We thank the reviewer for these comments. We checked all figures and updated the order of the references. The previous Figure 1 is now Supplementary Figure 1 and previous Figure 2 is now Figure 1. We furthermore revised the manuscript to address the Structural Points.

Reviewer #2 (Remarks to the Author):

This paper utilized a uniquely large, rich dataset (both in # of cells and # of samples) to investigate the relationship between PBMCs and neurodegenerative disease. The authors found that there were sex-specific changes in cell type proportion in a number of different diseases. Males with AD had higher levels of CD8+ T cells and females with MCI had higher levels of NKT-like cells and monocytes. They then built a random forest framework to try and identify a small number of genes that could be used to distinguish between cells from healthy donors vs. Patients with disease. They found that they were able to build 80% accurate models on cells and up to 90% accuracy on patients. They then conducted sex-stratified differential gene expression analyses between healthy controls and cases with neurodegenerative diseases. They found that the number of DEGs varied greatly between sexes across the same cell type, especially in AD. Through further analysis, they determined that the level of shared difference between diseases is common across sexes but that the signatures vary greatly between sexes. Finally, they wanted to compare the signal in PBMCs to the signal in the brain, so they took the dysregulated genes in PBMCs and looked for expression changes in cells from the brain. They found no correlation between gene expression in females and a slight anti-correlation in males. Overall, the authors seek to promote the utility of their large dataset and prompt others to consider the impact of sex when determining biomarkers of disease. The major strength of this work is the dataset generated, which will be a tremendous resource to the community. Several methodological weaknesses reduce enthusiasm regarding the main scientific analyses conducted and, thus, on the conclusions made from these analyses. Specifically, the random forest analysis is fraught with issues, noted in detail below, and does not seem to add much to the manuscript. There is also a missed opportunity to highlight the rich biomarker data that is available on their web tool (which is not really included in any of the analyses). Enriching the presented analyses and potentially expanding them to further explore correlations between cell types or gene expression could also add to the scientific value of this exciting dataset.

We appreciate the generally encouraging feedback. We agree that the size and power of the dataset is the major strength of our work. We however also understand the criticism, especially on the methodological side related to the single cell classification using random forests. Given the importance and weight of the comments in this regard below we decided to remove that aspect (previous Figure 4). We instead decided to focus the paper more on the strength mentioned by this reviewer by highlighting the webserver. Moreover, we also extended the analysis by using an extensive pathway analysis to strengthen the functional aspect of this work. Below, you can find responses to your comments.

Main Concerns:

- 1) Analyses are being run on raw cell proportions. It is prudent that the authors also report results from adjusted analyses of cell proportions, for example, using a regression model with adjustments for covariates (i.e., age, APOE e4 status (especially since it is sex split)), which is a common practice for the field and found in other, similar studies.

We thank the reviewer for this comment and added an analysis with scCODA, replacing some of the comparisons on raw cell-type proportions. We used Age and ApoE as confounding factors. We moved Figure 3a to the supplementary Figures and instead show the scCODA results in the new Figure 2 b and d and described the observed changes in the text: "These sex-dependent changes were confirmed using scCODA (Supplementary Data 2), which did not show significant changes in the cell-

type distributions (abs log2FC > 0.35, significant according to scCODA) when leaving out sex as a covariate (Supplementary Fig. 5c). Split by male and female patients, we were able to observe significant changes (Figure 2b), some of which were different dependent on the sex. In Parkinson's disease, CD8+ T cells and Plasma cells show a positive fold-change in males but a negative fold-change in females. Similarly, B cells are more abundant in females with PD-MCI but less abundant in males with PD-MCI. Although a sex-dependent change in B and NK cells of AD patients was indicated in the density embedding, scCODA did not show a significant difference. In the finer cell-annotation, we observed significant changes in only 3 cases in males and 9 cases in females (Figure 2c). The significant changes in AD, MCI and PD match the ones found in the female patients (Supplementary Fig. 5d). When comparing these findings with previously reported changes, some were previously described by others, such as the decreased proportion of CD8+ T cells in AD, and changes in B cells in PD (Figure 2c-d). Using scCODA on the finer Annotation, we found a lower proportion of Tfh cells, as previously reported (Figure 2e)."

- 2) The rationale for the random forest analysis is unclear. Among other issues, the fact that most of the genes identified are not differentially expressed raises some major red flags as to the validity of the method. In particular:

We appreciate this comment and take these concerns very seriously. After some consideration, we decided to remove the Random Forest part from the manuscript as explained above. Instead, we decided to focus more on the webserver and the added pathway analysis in the paper. Nonetheless we want to take the opportunity to share our view on the criticism by this reviewer, mentioning analyses we did, providing evidence that we are very concordant with his/her view.

- a) It is not clear what the objective of the model is.

- i) If the goal is to make predictions on disease status...

Indeed, the main focus was to demonstrate that we can predict disease based on single cells and in an improved manner after aggregation on the patient level.

I would love to see a comparison between these models and models using DEGs from the authors' other analyses (I think DEGs would do as well or better and make more sense scientifically as a starting point)

The authors could potentially use the same dropout method but use different DEG thresholds as a starting point

Most other models trying to do this are complex deep learning models with full data, not 2000 genes; what is the justification for using this subset? Is there precedent for this being a valid approach?

The random forest classification highlighted genes also showing up in the DEG analysis. Our concern was how well these results can be generalized with respect to the resampling splits. In our view an external data set is required to make this a fair comparison. This led us to the decision to omit the part and follow the suggestion of a more accurate benchmarking, also comparing with deep learning models at a later stage.

- ii) ii. If the goal is to identify genes that are different...

I think differential expression analysis is a much better way to do this; if this is not the case, then a very clear explanation and justification for this should be presented in the manuscript

We fully agree with this view.

- b) There are numerous missing methodological details, which diminishes transparency & reproducibility
- i) There is no mention of how feature importance is assessed. I assume it is a common python or R package, but it would be good to know which one as this is an important detail.
 - ii) I did not see any mention of hyperparameters for the random forest model (i.e., max depth, number of trees).
 - iii) Most ML methods use AUC or F1 as a metric rather than accuracy. If the authors choose not to use either of those, then a clear justification for why accuracy is a reasonable metric to use for evaluating the findings is required.

Although this part is not relevant anymore for the present submission, we share details of the analysis:

The feature importance was assessed using the Gini importance (mean decrease in impurity) which is inherent to random forests. The packages that were used were the ranger (version 0.14.1) package for the random forest classifiers and the caret (version 6.0-93) package for the backwards feature elimination. We used default parameters for the Random Forest classifier (num.trees=500, mtry = 44, max.depth = NULL). We chose accuracy since the number of cells from the two conditions were balanced in the independent test set. In our future work we will follow the advice and include AUC values and other common measures instead of relying on accuracy only.

- 3) The web server that they have set up looks like a great tool for exploring the data and provides a good community resource. The title of the paper also seems like it is designed to promote this feature. However, the results generated by this tool are not shown in the main body of the paper. I think that to show the utility of the tool and the additional biomarker and longitudinal data it houses, an analysis using the data highlighted within the tool should also be included in the manuscript.

We thank the reviewer for this comment. We agree with the reviewer and added new sections to highlight the data that is visualized on the webserver. See Results section: "The web-resource allows the visualization of gene-expression values (Extended data Fig. 7a-b). The direct access to the list of de-regulated genes and its visualization (Extended data Fig. 7c) allow fast hypothesis-testing. In addition to the scRNA-seq data used for the analysis shown in this manuscript, the webserver also allows researchers to access the longitudinal data, brain volume measurements and CSF-marker values." As the time-series, biomarker and brain data were highlighted by both Reviewer #1 and Reviewer #2, we added an additional supplementary figure (New Supplementary Figure 10 and 11), showing the features and limitations of that data in more detail.

Minor Concerns:

1. I would like to see a true Table 1 with a demographic breakdown rather than a series of graphs as provided in Figure 2. It is very difficult to derive meaningful information from the data as presented in Figure 2. It would also be more informative to provide statistical comparisons between each of the covariates between the sexes since this is one of the main contrasts presented in the manuscript.

We thank the reviewer for this comment and added the suggested table (Table 1) and included a statistical comparison between the ages in Supplementary Figure 2b.

2. The paper only shows sex-stratified results for cell proportion. It would also be helpful to provide a non-sex-stratified version so that it is possible to see if the signal is maintained but only statistically significant in smaller n populations (e.g., this was done for DEGs and there were not many overlapping hits).

We added this analysis to Supplementary Figure 5. We used both a t-test on the raw cell-type proportions (as in the previous Figure 3) and the scCODA corrected values.

“These sex-dependent changes were confirmed using scCODA (Supplementary Data 2), which did not show significant changes in the cell-type distributions ($\text{abs log}_2\text{FC} > 0.35$, significant according to scCODA) when leaving out sex as a covariate (Supplementary Fig. 5c). Split by male and female patients, we were able to observe significant changes (Figure 2b), some of which were different dependent on the sex.”

3. The Venn diagrams in Figure 2 are not very easy to decipher and many of the labels are hard to read. I would recommend the authors explore alternative strategies for presenting this information in a clearer way.

We apologies for this and changed the Venn diagram in the original Figure 2a (updated Figure 1a) into a table and the circle plot in the original Figure 2e (updated Figure 1d) into a bar-plot, labeling only cell-types that contribute more than 1.5% of the cells, hoping that this is easier to read.

4. When pathway analysis for DEGs is performed, only AD and PD-specific KEGG was run. It would be helpful to also include more general GO analysis, especially since the authors don't find any significant hits and openly admit that the pathways might be biased against sex-specific signals.

Thank you for this comment. We added the pathway analysis in the new Figure 4 and included a new section in the text: “This poses the question, which pathways are affected specifically in males and females. The results of a pathway analysis of the change in all cell-types and sexes (Supplementary Data 4) shows that the affected pathways in AD and in MCI are similar in both sexes (Figure 4a). In PD the pathways are more sex specific. The overall most frequently enriched pathway in AD is “SRP-dependent cotranslational protein targeting to membrane”. This is also the most frequently enriched pathway in females MCI and males with PD (Figure 4b). In PD, the “Asthma” pathway is most affected in females and in MCI “Oxidative phosphorylation” in males.

In PD, MCI and in males in PD, the pathways are similar across all cell-types (Supplementary Fig. 8). In females with AD, B cells show a similar cluster of pathways consisting of pathways related to junctions (adherens junction, cell-substrate junction, cell-substrate adherens junction) and pathways related to catabolic and biosynthetic processes (“amide biosynthetic process”, “aromatic compound catabolic process”, ...). In males, these pathways appear across all groups of cell-types.

Overall, there are 15 pathways shared between PD, MCI and AD in females, but 53 in males (Figure 4c,d). 114 pathways are shared between AD and PD in males and 24 in females (Figure 4d)."

We also added a comparison of the pathway analysis on the brain and the PBMC data in the updated Figure 5 (d,e,h and i) going into more detail into the potential mechanisms as suggested by Reviewer #3.

Other:

Most of the findings from the paper have to do with sex differences, however, the title makes no mention of this; modifying the title to highlight the main research findings would help readers interested in this topic find this resource

We thank the reviewer for this comment. We agree about the importance of the sex-specific effects in our study and changed the title to "A single-cell atlas to map sex-specific gene-expression changes in blood upon neurodegeneration" adequately communicates its importance in the manuscript.

Figure 1: it looks like some of the data points are duplicated (Wang x 2? Sirkis x 2?) and/or have incorrect values (e.g., does Sirkis really have only 100s of cells?).

The publications we refer to here are Wang, 2021 and Wang, 2022 as well as Sirkis, 2023 and Sirkis, 2024 (pre-print). The detailed information is available as Supplementary Table 1. To avoid confusion about these datasets we added the year of publication as a label to the plot. Sirkis contains transcription profiles of 182,000 PBMC cells, we corrected this in the manuscript and re-checked all the other values. As suggested by Reviewer #1, this plot was now moved into Supplementary Figure 1.

The references for Figure 1 data are incomplete

We apologize for this miscommunication. The complete reference are available in supplement table 2. We make this now clearer in the figure caption: "References to the datasets are available in Supplementary Table 1."

Figure 3c: the dots and the stars look very similar given the small size. Is there an alternative way to show this data?

We chose to use frames and dots and hope that this improves visibility now.

Extended data 7c-d: Y axis labels missing from plots

We revised the plots displayed in the webserver and added axis labels (See updated Supplementary Fig. 9).

Reviewer #3 (Remarks to the Author):

The topic of explore the peripheral immune system's role in neurodegenerative diseases is intriguing. While the results hold potential merit, the presence of several serious conceptual and technical flaws in methods and a lack of adequate validation on external datasets significantly reduce our support for publication in its current form.

We thank the reviewer for this valuable feedback. We worked to address the methodological challenges and shifted the focus away from the classification aspect in the previous figure 4, emphasizing the strength of the large data set, the web server and improved pathway analysis. Below you find detailed responses to each of your comments, along with the changes we made in the manuscript to address them.

The following are some considerations that could further improve the manuscript:

Major:

1. The demographic and clinical information of the sample population is not provided. The study should consider and discuss potential confounding factors such as age, medication use, other disease, lifestyle factors, etc., that might influence PBMC gene expression.

We apologize for missing details in this regard. We used matching age-groups to account for age as a confounding factor. We added a new Supplementary Figure 2b to show that there is no significant difference between age of the male and female patients. Several factors mentioned by this reviewer such as medication use and lifestyle factors of the patients were not available for this study. We thus focused on the analysis of the clinical data that was available, i.e. diagnosis, sex and changes in brain and CSF. We highlighted this limitation in the discussion:

“While a more specialized dataset with more samples (e.g. for AD) and additional demographic and clinical information (e.g. medication use, other disease, lifestyle factors, ...) would be necessary to study the exact mechanisms, this dataset allows us to generally study disease- and sex-specific changes in patients with neurodegenerative diseases.”

2. The study identifies changes in gene expression and cell type abundances, but the functional implications of these changes are not discussed. Further investigation and discussion about how these molecular changes translate to clinical symptoms or disease mechanisms would be valuable.

We thank the reviewer for this comment. We added a pathway analysis (new Figure 4) providing more insights into possible mechanisms. We still want to emphasize that, for a detailed explanation about the exact mechanisms, additional experiments would be necessary. We explain this now more clearly in the Discussion: “To study the exact mechanisms, a more specialized dataset with more samples (e.g. for AD) and additional demographic and clinical information (e.g. medication use, other disease, lifestyle factors, ...) would be necessary. Changes in the sub-cell types and potential sub-groups in the cell clusters furthermore need to be studied in more targeted experiments, as the number of cells in each of these clusters is limited.”

3. The manuscript does not provide a comparison of the random forest method with other machine learning techniques. It is important to ensure that this method is indeed the most suitable one for the specific data and objectives of the study.

Due to serious concerns from all three reviewers, we decided to remove the Random Forest model from the paper. We agree that a more rigorous evaluation and comparison to other statistical and machine learning approaches would be required and in addition that an unrelated test data set should be used to evaluate the performance.

4. In the method section, the author should explain how the hyperparameters in random forest were tuned and provide evidence that the selected parameters are indeed optimal.

Even though this point is not of relevance anymore we share details on the parameter selection: We opted for the default hyperparameters of random forests, since tuning of hyperparameters represents an increase in computational cost and we did not observe a significant improvement in performance when tuning the num.trees and mtry parameters.

5. It is difficult to understand why most of the features selected by the machine learning model are not differentially expressed. The authors should provide a more detailed explanation of the possible biological mechanisms underlying this observation.

We fully agree with this view, and it was part of why we removed the analyses. From our view the model did not generalize well enough, making the difference between the deregulated genes and the selected features. Please see also our response to point 2 of Reviewer #2.

6. The p-value adjusted method is not used in the correlation analysis. Given the large number of cells analyzed, adjusted p-values should be reported for the correlation analysis.

We apologize for this miscommunication. As for all other p-values, we used Benjamini-Hochberg to correct the values. We did state this in the "Statistics and reproducibility" section of the paper. To be clearer about this, we added the remark adjusted to all figures containing adjusted p-values and to all methods where it was applied.

7. In the manuscript, the authors acknowledge several previously generated single-cell datasets. They mention some consistencies and inconsistencies between their findings and those of other studies. However, the comparison can be improved by integrating in-house and public data. The incorporation of other datasets may increase statistical power and provide more insights from independent cohorts.

We thank the reviewer for this suggestion. We fully agree that integrating different datasets into our study would improve the statistical power. We try to apply this wherever possible, but the different experimental protocols make integrating the count data difficult and usually lead to less interpretable results. We thus decided to integrate the different studies at a later point by comparing our results with previously published ones (most not scRNA-seq, revised Figures 2e), to provide an overview of potential similarities and differences of our results to other datasets.

8. It is unclear how immune cells in PBMCs influence or are associated with disease initiation and development. Although the authors identified a few genes that are differentially expressed in both PBMCs and the brain, they did not demonstrate the function of these genes in a cell-type-specific manner, as the same genes may have

different functions in different cell types. Additionally, how PBMCs are linked to brain changes is unclear.

To address this comment, we added an extensive pathway analysis (see new Figure 5 f-h) and a comparison between blood and brain (see answers to 2 and 11). We found that the genes that are de-regulated in PBMCs and brain in females are enriched for immune system related pathways, indicating immune system related changes in both brain and blood.

9. The machine learning model designed to distinguish between healthy and diseased individuals needs further clarification and potential strengthening in the following aspects:
 - i. The authors need to clarify that the accuracy was calculated using cross-validation and not independent testing. Independent validations will better assess the model's accuracy.
 - ii. Does the abundance of cell types in PBMCs contribute to the prediction? Since the authors observed differences between healthy donors and diseased individuals, it is reasonable to hypothesize that cell abundance may have some predictive power. Will integrating cell abundance with gene signatures improve the prediction power?
 - iii. The best accuracy on the cell level was achieved with a model trained on B cells, and on the patient level, the highest accuracy was achieved using male NK cells. What are the functions of the signature genes identified by the machine learning model? How do these functions influence disease initiation and development?

As explained under section 3, we removed the feature selection using Random Forests from the manuscript.

10. Differentially Expressed Genes (DEG) Analysis:

What is the rationale behind the DEG analysis within cell types? Does the same cell type express different genes in different patients? As the authors already annotated cells at a high resolution, do they expect subpopulations within the current annotations? What are the differentially expressed genes within the same cell type, and what are their functions and impacts on disease development? This part needs further clarification.

- *We thank the reviewer for this comment. To highlight why it is important to perform the analysis on each cell-type, we added a new supplementary Figure 6 showing the similarities between changes in the gene-expression patterns. We were able to show that the changes were cell-type specific for T and B cells, although the differences in the sub-cell types are rather minor.*
“When comparing the gene-expression changes in the different cell-types, we observed that the changes sub-cell types from the same cell type show a high similarity (Supplementary Fig. 6).”
- *The annotation is already on a relatively high resolution for this dataset (see Supplementary Figure 4). Annotating the dataset even finer would not be*

recommended. That does not mean that there are no potential sub-populations of cells in AD, PD or MCI, but other, more targeted experiments would be necessary to detect them and study the changes observed in them. We state this more clearly in the Discussion section: “Changes in the sub-cell types and potential sub-groups in the cell clusters furthermore need to be studied in more targeted experiments, as the number of cells in each of these clusters is limited.”

To go into more detail about the function and role of the differentially expressed genes, we performed a pathway analysis based on the de-regulation in the different cell-types in the new Figure 4 and added a new paragraph going into more detail here: “This poses the question, which pathways are affected specifically in males and females. The results of a pathway analysis of the change in all cell-types and sexes (Supplementary Data 4) shows that the affected pathways in AD and in MCI are similar in both sexes (Figure 4a). In PD the pathways are more sex specific. The overall most frequently enriched pathway in AD is “SRP-dependent cotranslational protein targeting to membrane”. This is also the most frequently enriched pathway in females MCI and males with PD (Figure 4b). In PD, the “Asthma” pathway is most affected in females and in MCI “Oxidative phosphorylation” in males.”

11. PBMC and Brain Disease Signatures:

In the section “PBMC disease signatures are inversely correlated with changes in neural cells,” the authors integrate data generated using both brain and PBMCs and aim to show how changes in PBMCs are associated with changes in the brain in disease. Several questions need to be addressed in this section:

- i. Are there any shared cell types between PBMCs and the brain in patients?
- ii. How do changes in PBMCs influence changes in the brain or vice versa?
- iii. What are the genes (43 genes in males and 8 genes in females) that are deregulated in both PBMCs and the brain, and what are their functions?
- iv. The mechanism by which the brain interacts with PBMCs in patients remains unclear and needs further elucidation.

We thank the reviewer for this valuable feedback. We enriched our analysis with additional pathway analysis and comparisons of the de-regulation of genes involved in cell-cell signaling, providing possible mechanisms for how the blood and brain might interact.

- i. Are there any shared cell types between PBMCs and the brain in patients?

There are no overlapping cell-type clusters in PBMCs and brain such as T and B cells. Unfortunately, these contribute only a few hundred cells to the brain dataset and thus don't appear in the results of the DEG analysis. We state this more clearly in the Results: “As there are not enough cells from the same cell-type in both brain and blood, we rely on overlaps of gene-expression patterns and pathways when looking for similar patterns in neurodegenerative diseases.”

There are other clusters with immune cells in the brain such as Microglia, but these resident immune cells and no peripheral immune cells and thus can show different transcriptomic patterns.

- ii. How do changes in PBMCs influence changes in the brain or vice versa?

To answer this question, we performed two additional analyses:

- i. *The first analysis (new Figures 5 e,g) tried to identify signaling pathways between the blood and brain that are influenced by changes in the blood.*

- ii. *The second one (new Figure i) aimed at detecting systemic changes in Alzheimer's patients that can thus be detected both in the blood and the brain by performing and comparing additional pathway analysis on both brain and blood.*
- iii. What are the genes (43 genes in males and 8 genes in females) that are deregulated in both PBMCs and the brain, and what are their functions?
We performed an over representation analysis (ORA) with the genes in question and added them to the Results (updated Figure 5d): "The 43 genes in males were enriched for pathways related to the Immune system regulation and the membrane (Figure 5d,e). For the 8 female genes, no significantly enriched or depleted pathways were found."

The mechanism by which the brain interacts with PBMCs in patients remains unclear and needs further elucidation.

We added an additional pathway analysis and checked for de-regulated signaling pathways and added Figure 5f-h to further go into detail with possible mechanisms. Of course, further experimental validation on respective mechanisms will finally be required and we mention this aspect in the discussion: "To study the exact mechanisms, a more specialized dataset with more samples (e.g. for AD) and additional demographic and clinical information (e.g. medication use, other disease, lifestyle factors, ...) would be necessary."

Reviewer #3 (Remarks on code availability):

The code repository includes scripts for generating all the figures in the manuscript as well as most of the code used for data processing.

Code availability is of high importance, and we of course happily share our code for transparency reasons and reproducibility.

REVIEWER COMMENTS

Reviewer #1:

I would like to thank the authors for this in depth revisions of the manuscript. My concerns have been adequately addressed! I see a very valuable and sound manuscript that merits it consideration for this journal.

We greatly appreciate the very positive evaluation of our manuscript.

Reviewer #2:

The authors addressed all of the main feedback from the initial review and replaced the most problematic analysis with a new, more compelling one. The authors' updates to the statistical methods are more robust and this is greatly appreciated.

We thank the reviewer for this very positive feedback.

However, there remain some minor issues that would be ideal to address:

1) While the authors created a Table 1 and did some age comparisons between sex in each disease state, the significance from this test should also be included in the table. For full transparency, it would be helpful to provide a comparison across all covariates between sexes in the table as well (including statistical comparison results).

We thank the reviewer for this clarification. We included the information about the significance between different covariates (race, age) in male and female in table 1.

2) While it was very helpful that the authors added the years to the figure showing other PBMC datasets and included a separate reference table, one of the datapoints still shows only 100 cells (which is incorrect, as noted by the authors themselves in the response to reviewers document and throws off the scale of the graph. Please ensure all values are accurately updated in this figure.

We excuse for this mistake. We updated the figure and re-checked the publications included.

Once these minor issues are addressed, the manuscript will represent a compelling body of work worthy of publication.

Reviewer #3:

The authors have made some adjustments in response to our comments. However, we think most of our concerns remain unaddressed.

Demographic and Clinical Information: While the authors acknowledged the lack of detailed clinical data, such as medication use and lifestyle factors, their response is unsatisfactory. Merely controlling for age and sex without considering other potential confounding factors significantly undermines the validity of their conclusions. The absence of these important details should not be dismissed, and this aspect was not addressed with the rigor required.

Although we agree that adding more clinical data to the dataset would be interesting, we cannot add additional information about the patients at this point. We still think that our conclusions are valid, as considering medication use and lifestyle factors in the analysis of single-cell data is not typically part of the analysis of single-cell data analysis in this research area. We added an overview of Alzheimer's studies and which confounding factors they include and consider during the analysis in Table 1 in this document. For the PBMC studies in Alzheimer's research, no study included more information about lifestyle and medication, and none considered any confounding factor despite sex and age. And even though brain studies frequently include those meta-data, addressing them and considering them as confounding factors is usually not included in the manuscripts. Thus, this approach is state-of-the-art methodology in Alzheimer's research.

Functional Implications of Changes: The added pathway analysis is a step forward, but it feels more like an attempt to patch up the original concerns than a fundamental solution. The explanation remains superficial, and the authors fail to adequately explore the broader clinical or biological implications of the observed changes. More in-depth exploration of how these findings relate to disease mechanisms is still lacking.

Investigating the functional mechanisms and providing and validating the functional explanations of the observations that we see in this dataset does in our eyes go beyond what we can address in the scope of this manuscript. We agree that this is an important area for future investigation, but it is outside of the scope of the current work. To provide some insights

into possible biological implications, we added a section to the Discussion providing an overview about previously published insights:

- *For the cell-type composition:* “Changes in B cells and Monocytes in Alzheimer’s disease were inconsistent across male and female patients. It has been previously shown that depletion of B cells from mice led to cognitive deficits and an increase of A β plaques [58]. B cells also play an important role in the regulation of local immune responses and might thus indicate inflammatory states in Alzheimer’s. The increased levels of monocytes in AD can furthermore be a sign of chronic inflammation and may have a negative effect on structures like the blood brain barrier (BBB) [59].”
- *For the pathways that were observed:* “Changes in B cells and Monocytes in Alzheimer’s disease were inconsistent across male and female patients. It has been previously shown that depletion of B cells from mice led to cognitive deficits and an increase of A β plaques [58]. B cells also play an important role in the regulation of local immune responses and might thus indicate inflammatory states in Alzheimer’s. The increased levels of monocytes in AD can furthermore be a sign of chronic inflammation and may have a negative effect on structures like the blood brain barrier (BBB) [59].”
- *And for the link between blood and brain:* “Changes in B cells and Monocytes in Alzheimer’s disease were inconsistent across male and female patients. It has been previously shown that depletion of B cells from mice led to cognitive deficits and an increase of A β plaques [58]. B cells also play an important role in the regulation of local immune responses and might thus indicate inflammatory states in Alzheimer’s. The increased levels of monocytes in AD can furthermore be a sign of chronic inflammation and may have a negative effect on structures like the blood brain barrier (BBB) [59].”

Random Forest Analysis: Simply removing the Random Forest model is an evasive move. Instead of answering legitimate questions about their methodology, they have sidestepped the issue altogether. This approach does not address the core problem of methodological robustness. The absence of any comparison to other machine learning models or justification for their initial approach leaves a significant gap in the study’s reliability.

During the Revision process, we removed the Random Forest Part as we shared the reviewers concerns and as changing from accuracy to F1 score (comment of Reviewer #2) revealed problems with the training. As we were not able to fully resolve this issue and found that the

classification of Alzheimer's data is probably too complex for this approach, we decided to remove it from the paper. This was also positively highlighted by reviewer #2: "The authors addressed all of the main feedback from the initial review and replaced the most problematic analysis with a new, more compelling one.". In our eyes, the Random Forest based analysis at this point did not add any value to the described dataset.

We anyways agree with the reviewer that publishing these negative results also benefits the research in this area and thus decided to include a new Supplementary Information Document describing the tests and results that we got from the Random Forest based feature selection. We added a new paragraph to the manuscript:

"To identify sex- and disease-specific signature, a machine-learning model was applied to the dataset (Supplementary Notes 1, Supplementary Data 3). Testing out a random forest-based feature selection and three other machine-learning methods, we concluded that the application on an Alzheimer's disease dataset is too complex and would require a larger dataset (ideally with different studies) to reliably predict disease-specific features across patients."

PBMC and Brain Disease Signatures: The additional pathway analyses, while informative, do not resolve the deeper concern about the connection between PBMC and brain data. The authors fail to address how the changes in PBMCs might relate to brain disease mechanisms in a meaningful way. Their reliance on pathway overlaps does not provide the necessary mechanistic insight, and their explanation remains overly general.

We agree that more in-depth explanations about the exact mechanisms of how the PBMCs interact with the brain would be beneficial. We discussed the current research in the Introduction and Discussion:

"Indeed, accumulating evidence points to a partial breakdown of the BBB with age and during age-related diseases, suggesting a still underestimated role of the peripheral immune system in neurodegenerative conditions, potentially through information exchange between these otherwise restrictively isolated physiological environments [19-22]. These findings raise questions about how and at which point during the disease progression the peripheral immune system is implicated, especially in the context of driving systemic inflammation [23-26]."

“While the exact relationship between the immune-system and the brain remains unknown, this study highlights the importance to consider possible influences of the immune system and the sex on the development of these diseases.”

Unfortunately, finding the exact mechanisms would require years of further research and validation experiments and are out of scope for this study. Instead, we tried to provide more information about the changes that can be observed in different regions of the brain and how they relate to changes in PBMCs. For this, we added an additional dataset for the comparison between PBMCs and brain that contains multiple different brain regions (see updated Figure 5). We furthermore added a section describing if and in which context the overlapping genes were previously found in Alzheimer’s. “To further validate these findings and to see if the expression changes with different brain-regions, we tested for the general overlap of differentially expressed genes (abs. log₂ Fold-change > 0.5 and adj. P-value < 0.05) in this PBMC dataset, the ZEBRA dataset (21 studies) and the ROSMAP dataset (4 studies; 6 brain regions). Of note, two studies from the prefrontal cortex samples of the ROSMAP dataset are a subset of the ZEBRA dataset and thus not independent. The other brain-regions are independent between the datasets. We found an overlap of 32 genes in the male patients and 8 genes in the female patients. 24 out of the 32 male genes and 5 out of the 8 female genes have been previously reported in the context of Alzheimer’s disease, most in the context of the immune system, blood-brain barrier, Astrocytes and Microglia (Supplementary Table 3, Figure 5 d-e).”

We furthermore added a section in the discussion that describes these changes in the context of possible mechanisms of how and why these changes are relevant for disease development: “To further validate these findings and to see if the expression changes with different brain-regions, we tested for the general overlap of differentially expressed genes (abs. log₂ Fold-change > 0.5 and adj. P-value < 0.05) in this PBMC dataset, the ZEBRA dataset (21 studies) and the ROSMAP dataset (4 studies; 6 brain regions). Of note, two studies from the prefrontal cortex samples of the ROSMAP dataset are a subset of the ZEBRA dataset and thus not independent. The other brain-regions are independent between the datasets. We found an overlap of 32 genes in the male patients and 8 genes in the female patients. 24 out of the 32 male genes and 5 out of the 8 female genes have been previously reported in the context of Alzheimer’s disease, most in the context of the immune system, blood-brain barrier, Astrocytes and Microglia (Figure 5 d-e).”

Hyperparameter Tuning for Random Forest: Although the Random Forest model has been removed, their explanation of hyperparameter tuning is irrelevant now. The authors should have demonstrated why their approach was valid in the first place rather than simply avoiding the issue.

We thank the reviewer for this comment and added a Supplementary Note to the Supplementary Information Document that explains why we removed it from the main manuscript.

Machine Learning Model Explanation: Again, the removal of the Random Forest analysis is an unsatisfactory way of dealing with this concern. Instead of improving the model or providing clear justification for their method, they have taken the easy way out. This undermines the overall rigor of their study.

See answer to the previous comment.

In summary, we think the rebuttal falls short of adequately addressing the core concerns raised. Instead of improving their methodology or providing detailed explanations, the authors have opted to remove the problematic aspects of their analysis without properly defending or revising their approach. This is not an acceptable solution, and many of the fundamental questions remain unanswered. We, therefore, recommend not accepting this paper.

Table 1: Availability and usage of meta-data information in single-cell Alzheimer's studies. (*) study is still pre-print and the information is not yet available online.

Author	DOI	Tissue	Data available	Data examined						year
				Age	Sex	lifestyle	medication	race	education	
Mathys et al.	https://doi.org/10.1038/s41586-019-1195-2	Brain	yes	yes	yes	no	no	no	no	2019
Mathys et al.	https://doi.org/10.1016/j.cell.2023.08.039	Brain	yes	yes	yes	no	no	yes	no	2023
Mathys et al.	https://doi.org/10.1038/s41586-024-07606-7	Brain	yes	yes	yes	no	no	no	no	2024
Fujita et al.	https://doi.org/10.1038/s41588-024-01685-y	Brain	yes	yes	yes	no	no	yes	no	2024
Green et al.	https://doi.org/10.1038/s41586-024-07871-6	Brain	yes	yes	yes	no	no	yes	yes	2024
Blanchard et al.	https://doi.org/10.1038/s41586-022-05439-w	Brain	yes	yes	yes	no	no	no	no	2022
Gabitto et al.	https://doi.org/10.1038/s41593-024-01774-5	Brain	yes	yes	yes	no	no	yes	yes	2024
Sirkis et al. *	https://doi.org/10.1101/2023.09.26.559634	PBMCs	yes *	yes	yes	no	no	no	no	2024
Xiong et al.	https://doi.org/10.1038/s12276-021-00714-8	PBMCs	yes	yes	yes	no	no	no	no	2021
Xu et al.	https://doi.org/10.3389/fimmu.2021.645666	PBMCs	yes (age, sex)	yes	yes	no	no	no	no	2021

Answers to Reviewers:

Reviewer #2:

Overall, the main strength of this study remains the incredible data resource and exploration portal that have been generated, which will be of great interest to a broad audience of researchers. The findings presented highlight the importance of sex differences in biology and omics research and will hopefully seed future work that can further explore sex-differences using this and other data resources. The authors have largely addressed most of the prior comments and suggestions, and the methods and results presented are satisfactory. There are a few minor additional items that would be worth addressing to enhance the presentation of results and accuracy of the study prior to final publication, though addressing these would not preclude the manuscript's consideration for publication.

We thank the reviewer for this overall positive feedback.

Minor comments:

In the comparison of males and females in each disease by number of deregulated genes in Figure 3a, it might be easier to spot the difference if each disease was lined up next to the other.

We reordered the columns in Figure 3a to make it easier to compare.

The section discussing the failure of the Random forest model would be better placed in the Discussion section rather than in its current location.

We thank the reviewer for this suggestion and moved this section to the discussion.

Please ensure the number of patients in ROSMAP is accurately reported.

We excuse for this confusion. We checked the data again and it fits to our dataset. We used three studies including Blanchard 2022¹, Mathys 2023², Fujita 2024³, and Mathys 2024⁴.

While the data sets seem to be only partially overlapping (Mathys 2023, Mathys 2024) we also found an overlap to Blanchard 2022 which was confirmed by the authors. Moreover, we figured out a substantial overlap of 8 Patients, between Blanchard 2022 and Mathys 2019⁵ including 10x Chromium v2 samples, this has been confirmed by the authors as well⁶. The data was directly realigned starting from the reads deposited on

Synapse, including one additional donor not used in Blanchard 2022 study but deposited under Synapse **syn2580853**, which leads to 25 remaining donors from this folder. This leaves our analysis with a cleaned-up data set with 603 donors in total from the ROSMAP cohort across the four studies.

- 1 *Blanchard, J. W. et al. APOE4 impairs myelination via cholesterol dysregulation in oligodendrocytes. Nature* **611**, 769-779 (2022). <https://doi.org:10.1038/s41586-022-05439-w>
- 2 *Mathys, H. et al. Single-cell atlas reveals correlates of high cognitive function, dementia, and resilience to Alzheimer's disease pathology. Cell* **186**, 4365-4385.e4327 (2023). <https://doi.org:https://doi.org/10.1016/j.cell.2023.08.039>
- 3 *Fujita, M. et al. Cell subtype-specific effects of genetic variation in the Alzheimer's disease brain. Nature Genetics* **56**, 605-614 (2024). <https://doi.org:10.1038/s41588-024-01685-y>
- 4 *Mathys, H. et al. Single-cell multiregion dissection of Alzheimer's disease. Nature* **632**, 858-868 (2024). <https://doi.org:10.1038/s41586-024-07606-7>
- 5 *Mathys, H. et al. Single-cell transcriptomic analysis of Alzheimer's disease. Nature* **570**, 332-337 (2019).
- 6 *Blanchard, J. W. et al. Author Correction: APOE4 impairs myelination via cholesterol dysregulation in oligodendrocytes. Nature* **636**, E9-E9 (2024). <https://doi.org:10.1038/s41586-024-08434-5>

Reviewer #3:

We appreciate the authors' efforts in addressing our concerns. We believe the dataset has the potential to benefit the community, and the findings offer valuable insights for Alzheimer's research.

We thank the reviewer for this positive feedback.